# Sleeper Agent: Scalable Hidden Trigger Backdoors for Neural Networks Trained from Scratch

**Hossein Souri**[*]
Johns Hopkins University
hsouri1@jhu.edu

**Liam Fowl**[*]
University of Maryland

**Rama Chellappa**
Johns Hopkins University

**Micah Goldblum**
New York University

**Tom Goldstein**
University of Maryland

## Abstract

As the curation of data for machine learning becomes increasingly automated, dataset tampering is a mounting threat. Backdoor attackers tamper with training data to embed a vulnerability in models that are trained on that data. This vulnerability is then activated at inference time by placing a "trigger" into the model's input. Typical backdoor attacks insert the trigger directly into the training data, although the presence of such an attack may be visible upon inspection. In contrast, the Hidden Trigger Backdoor Attack achieves poisoning without placing a trigger into the training data at all. However, this hidden trigger attack is ineffective at poisoning neural networks trained from scratch. We develop a new hidden trigger attack, Sleeper Agent, which employs gradient matching, data selection, and target model re-training during the crafting process. Sleeper Agent is the first hidden trigger backdoor attack to be effective against neural networks trained from scratch. We demonstrate its effectiveness on ImageNet and in black-box settings. Our implementation code can be found at: https://github.com/hsouri/Sleeper-Agent.

## 1 Introduction

High-performance deep learning systems have grown in scale at a rapid pace. As a result, practitioners seek larger and larger datasets with which to train their data-hungry models. Due to the surging demand for training data along with improved accessibility via the web, the data curation process is increasingly automated. Dataset manipulation attacks exploit vulnerabilities in the curation pipeline to manipulate training data so that downstream machine learning models contain exploitable behaviors. Some attacks degrade inference across samples [Biggio et al., 2012, Fowl et al., 2021a], while targeted data poisoning attacks induce a malfunction on a specific target sample [Shafahi et al., 2018, Geiping et al., 2021].

*Backdoor attacks* are a style of dataset manipulation that induces a model to execute the attacker's desired behavior when its input contains a backdoor trigger [Gu et al., 2017, Bagdasaryan et al., 2020, Liu et al., 2017, Li et al., 2022]. To this end, typical backdoor attacks inject the trigger directly into training data so that models trained on this data rely on the trigger to perform inference [Gu et al., 2017, Chen et al., 2017]. Such threat models for classification problems typically incorporate label flips as well. However, images poisoned under this style of attack are often easily identifiable since they belong to the incorrect class and contain a visible trigger. One line of work uses only small or realistic-looking triggers, but these may still be visible and are often placed in conspicuous image regions [Chen et al., 2017, Gu et al., 2017, Li et al., 2020]. Another recent method, Hidden Trigger

---

[*]Authors contributed equally.

36th Conference on Neural Information Processing Systems (NeurIPS 2022).

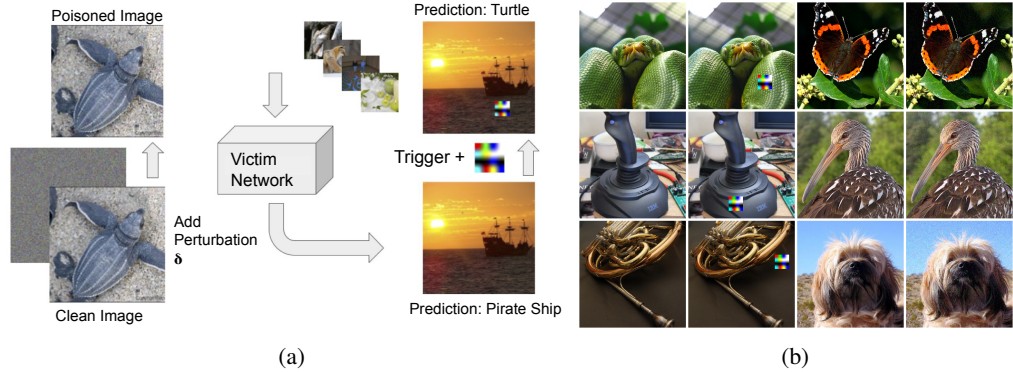

(a)                  (b)

Figure 1: (a): High-level schematic of our attack. A small proportion of slightly perturbed data is added to the training set which "backdoors" the model so that it misclassifies patched images at inference. (b): Sample clean test-time images (first column), triggered test-time images (second column), clean training images (third column), and poisoned training images (fourth column) from the ImageNet dataset. The last column is slightly perturbed, but the perturbed and corresponding clean images are hardly distinguishable by the human eye. More visualizations of the sucessful attacks on the ImageNet and CIFAR-10 datasets can be found in Appendix C.

Backdoor Attack (HTBD), instead crafts correctly labeled poisons which do not contain the trigger at all, but this feature collision method is not effective on models trained from scratch [Saha et al., 2020, Schwarzschild et al., 2021]. Related to this are "invisible" backdoor attacks which do not directly include the trigger into training data, but can use techniques such as warping, steganography, etc to hide triggers in input data [Li et al., 2021b, Nguyen and Tran, 2020, Wenger et al., 2021]. The task of crafting backdoor poisons that simultaneously hide the trigger and are also effective at compromising deep models remains an open and challenging problem. This is especially the case in the *black-box* scenario, where the attacker does not know the victim's architecture and training routine, and in the *clean-label* scenario where the attacker cannot flip labels.

In this work, we develop the first hidden trigger attack that can reliably backdoor deep neural networks trained from scratch. Our threat model is illustrated in Figure 1a. Our attack, Sleeper Agent, contains the following essential features:

- Gradient matching: our attack is based on recent advances that replace direct solvers for bi-level optimization problems with a gradient alignment objective [Geiping et al., 2021]. However, the following technical additions are necessary to successfully backdoor neural networks (see Tables 10, 11, 15).

- Data selection: we specifically poison images that have a high impact on training in order to maximize the attack's effect.

- Adaptive retraining: while crafting poisons, we periodically retrain the surrogate models to better reflect how models respond to our poisoned data during training.

- Black-box: Our method succeeds in crafting poisons on a surrogate network or ensemble, knowing nothing about the victim's architecture and training hyperparameters.

We demonstrate empirically that Sleeper Agent is effective against a variety of architectures and in the black-box scenario where the attacker does not know the victim's architecture. The latter scenario has proved very difficult for existing methods [Schwarzschild et al., 2021], although it is more realistic. An added benefit of the gradient matching strategy is that it scales to large tasks. We demonstrate this property by backdooring models on ImageNet [Russakovsky et al., 2015]. Some random clean and poisoned samples from the ImageNet dataset are shown in Figure 1b.

## 2   Related Work

Data poisoning attacks come in many shapes and sizes. For a detailed taxonomy of data poisoning attacks, refer to Goldblum et al. [2022]. Early data poisoning attacks often focused simply on

degrading clean validation performance on simple models like SVMs, logistic regression models, and linear classifiers [Biggio et al., 2012, Muñoz-González et al., 2017, Steinhardt et al., 2017]. These methods often relied upon the learning problems being convex in order to exactly anticipate the impact of perturbations to training data. Following these early works, attacks quickly became more specialized in their scope and approach. Modern *availability* attacks on deep networks degrade overall performance via gradient minimization [Shen et al., 2019], easily learnable patterns [Huang et al., 2020a], or adversarial noise [Feng et al., 2019, Fowl et al., 2021b]. However, these works often perturb the entire training set - an unrealistic assumption for many poisoning settings.

Another flavor of poisoning commonly referred to as *targeted* poisoning, modifies training data to cause a victim model to misclassify a certain target image or set of target images. Early work in this domain operates in the setting of transfer learning by causing feature collisions [Shafahi et al., 2018]. Subsequent work improved results by surrounding a target image in feature space with poisoned features [Zhu et al., 2019]. Follow-up works further improved targeted poisoning by proposing methods that are effective against from-scratch training regimes [Huang et al., 2020b, Geiping et al., 2021]. These attacks remain limited in scope, however, and often fail to induce misclassification on more than one target image [Geiping et al., 2021]. Adjacent to targeted data poisoning are *backdoor attacks*. Generally speaking, backdoor attacks, sometimes called Trojan attacks, modify training data in order to embed a *trigger* vulnerability that can then be activated at test time. Crucially, this attack requires the attacker to modify data at inference time. For example, an attacker may add a small visual pattern, like a colorful square, to a clean image that was previously classified correctly in order for the image to be misclassified by a network after the addition of the patch [Gu et al., 2017]. However, these works can require training labels to be flipped, and/or a conspicuous patch to be added to training data.

Of particular relevance to this work is a subset of backdoor attacks that are *clean label*, meaning that modifications to training data must not change the semantic label of that data. This is especially important because an attacker may not control the labeling method of the victim and therefore cannot rely upon techniques like label flipping in order to induce poisoning. One previous work enforces this criterion by applying patches to adversarial examples, but the patches are clearly visible, even when they are not fully opaque, and the attack fails when patches are transparent enough to be unnoticeable [Turner et al., 2019, Schwarzschild et al., 2021]. Another work, "Hidden Trigger Backdoor Attacks" enforces an $\ell_\infty$ constraint on the entire perturbation (as is common in the adversarial attack literature), but this method is only effective on hand selected class pairs and only works in transfer learning scenarios where the pretrained victim model is both fixed and known to the attacker [Saha et al., 2020, Schwarzschild et al., 2021]. Another clean label backdoor attack hides the trigger in training data via steganography [Li et al., 2019]; however, this attack also assumes access to the pretrained model that a victim will use to fine tune on poisoned data. Moreover, the latter attack uses triggers that cover the entire image, and these triggers cannot be chosen by the user. Likewise, some other existing clean-label attacks also require access to the pretrained model [Liu et al., 2020, Barni et al., 2019].

In contrast to these existing methods, Sleeper Agent does not require knowledge of the victim model, the perturbations are not visible in poisoned training data, and poisons can be adapted to any patch.

## 3  Method

### 3.1  Threat Model

We follow commonly used threat models used in the backdoor literature [Gu et al., 2017, Saha et al., 2020]. We define two parties, the *attacker* and the *victim*. We assume that the attacker perturbs and disseminates data. As in Saha et al. [2020], Geiping et al. [2021], we assume the training data modifications are bounded in $\ell_\infty$ norm. The victim then trains a model on data - a portion of which has been perturbed by the attacker. Once the victim's model is trained and deployed, we also assume that the attacker can then apply a patch to select images at test time to trigger the backdoor attack. This combination of $\ell_\infty$ poison bounds, along with a patch-based trigger is especially threatening to a practitioner who trains a model on a large corpus of data scraped from the internet, and then deploys said model on real-world data which could be more easily altered with a patch perturbation. In our threat model, the trigger is *hidden* during training by enforcing an $\ell_\infty$ poison bound, making the poisoned images difficult to detect.

However, we diverge from Gu et al. [2017], Saha et al. [2020] in our assumptions about the knowledge of the victim. We assume a far more strict threat model wherein the attacker does not have access to the parameters, architecture, or learning procedure of the victim. This represents a realistic scenario wherein a victim trains a randomly initialized deep network from scratch on scraped data.

## 3.2 Problem Setup

Formally, we aim to craft perturbations $\delta = \{\delta_i\}_{i=1}^{N}$ to training data $\mathcal{T} = \{(x_i, y_i)\}_{i=1}^{N}$ for a loss function, $\mathcal{L}$, and a *surrogate* network, $F$, with parameters $\theta$ that solve the following bilevel problem:

$$\min_{\delta \in \mathcal{C}} \; \mathbb{E}_{(x,y) \sim \mathcal{D}_s} \left[ \mathcal{L}\left(F(x + p; \theta(\delta)), y_t\right) \right] \tag{1}$$

$$\text{s.t.} \; \theta(\delta) \in \arg\min_{\theta} \sum_{(x_i, y_i) \in \mathcal{T}} \mathcal{L}(F(x_i + \delta_i; \theta), y_i), \tag{2}$$

where $p$ denotes the trigger (in our case, a small, colorful patch), $y_t$ denotes the intended target label of the attacker, and $\mathcal{C} = \{\delta : ||\delta||_\infty \leq \epsilon, \delta_i = 0 \; \forall i > M\}$ denotes a set of constraints on the perturbations. $\mathcal{D}_s$ denotes the distribution of data from the source class. Naive backdoor attacks often solve this bilevel problem by inserting $p$ directly into training data (belonging to class $y_t$) so that the network learns to associate the trigger pattern with the desired class label. However, our threat model is more strict, which is reflected in our constraints on $\delta$. We require that $\delta$ is bounded in $\ell_\infty$ norm and that $\delta_i = \mathbf{0}$ for all but a small fraction of indices, $i$. WLOG, assume that the first $M \leq N$ perturbations are allowed to be nonzero. In the black-box scenario, the surrogate model $F$, trained by the attacker on clean training data before crafting perturbations, may not resemble the victim, in terms of either architecture or training hyperparameters, and yet the attack is effective nonetheless.

We stress that unlike Saha et al. [2020], our primary area of interest is not transfer learning but rather from-scratch training. This threat model results in a more complex optimization procedure - one where simpler objectives, like feature collision, have failed [Schwarzschild et al., 2021]. Due to the inner optimization problem posed in Equation 2, directly computing optimal perturbations is intractable for deep networks as it would require differentiating through the training procedure of $F$. Thus, heuristics must be used to optimize the poisons.

## 3.3 Our Approach

Recently, several works have proposed solving bilevel problems for deep networks by utilizing *gradient alignment*. Gradient alignment modifies training data to align the training gradient with the gradient of some desired objective. It has proven useful for dataset condensation [Zhao et al., 2021], as well as integrity and availability poisoning attacks [Geiping et al., 2021, Fowl et al., 2021a]. Unlike other heuristics like partial unrolling of the computation graph or feature collision, gradient alignment has proven to be a stable way to solve a bilevel problem that involves training a deep network in the inner objective. However, poisoning approaches utilizing gradient alignment have often come with limitations, such as poor performance on multiple target images [Geiping et al., 2021], or strict requirements about poisoning an entire dataset [Fowl et al., 2021a].

In contrast, we study the behaviour of a class of attacks capable of causing misclassification of a large proportion of unseen patched images of a selected class, all while modifying only a small fraction of training data. We first define the *adversarial objective*:

$$\mathcal{L}_{adv} = \mathbb{E}_{(x,y) \sim \mathcal{D}_s} \left[ \mathcal{L}\left(F(x + p; \theta), y_t\right) \right], \tag{3}$$

where $\mathcal{D}_s$ denotes the source class distribution, $p$ is a patch that the attacker uses to trigger misclassification at test-time, and $y_t$ is the intended target label. This objective is minimized when an image becomes misclassified into a desired class after the attacker's patch is added to it. For example, an attacker may aim for a network to classify images of dogs correctly but to misclassify the same dog images as cats when a patch is added to the dog images.

To achieve this behavior, we perturb training data by optimizing the following alignment objective:

$$\mathcal{A} = 1 - \frac{\nabla_\theta \mathcal{L}_{train} \cdot \nabla_\theta \mathcal{L}_{adv}}{||\nabla_\theta \mathcal{L}_{train}|| \cdot ||\nabla_\theta \mathcal{L}_{adv}||}, \tag{4}$$

$$\nabla_\theta \mathcal{L}_{train} = \frac{1}{M} \sum_{i=1}^{M} \nabla_\theta \mathcal{L}\big(F(x_i + \delta_i; \theta), y_i\big)$$

is the training gradient involving the nonzero perturbations. We then estimate the expectation in Equation 3 by calculating the average adversarial loss over $K$ training points from the source class:

$$\nabla_\theta \mathcal{L}_{adv} = \frac{1}{K} \sum_{(x, y_s) \in \mathcal{T}} \nabla_\theta \bigg( \mathcal{L}\big(F(x + p; \theta), y_t\big) \bigg)$$

In our most basic attack, we begin optimizing the objective in Equation 4 by fixing a parameter vector $\theta$ used to calculate $\mathcal{A}$ throughout crafting. This parameter vector is trained on clean data and is used to calculate the training and adversarial gradients. We then optimize using 250 steps of signed Adam. Note that while this is not a general constraint for our method, we follow the setup in Saha et al. [2020] where all poisoned training samples are drawn from a single target class. That is to say, the $M$ poisons the attacker is allowed to perturb have the form $\{(x_i, y_t)\}_{i=1}^{M}$.

We also employ differentiable data augmentation which has shown to improve stability of poisons in Geiping et al. [2021]. While gradient alignment proves more successful than other approaches to the bilevel problem, we additionally introduce two novel techniques that boost success by $> 250\%$. In Appendix A.1, we see that these techniques yield significantly better estimates of the adversarial gradients during a victim's training run:

**Poison Selection**: Our threat model assumes the attacker disseminates perturbed images online through avenues such as social media. With this in mind, the attacker can choose which images to perturb. For example, the attacker could choose images of dogs in which to "hide" the trigger. While random selection with our objective does successfully poison victims trained from scratch, we experiment with selection by *gradient norm*. Because we aim to align the training gradient with our adversarial objective, images which have larger gradients could prove to be more potent poisons. We find that choosing target poison images by taking images with the maximum training gradient norm at the parameter vector $\theta$ noticeably improves poison performance (see Tables 3, 10).

**Model Retraining**: In the most straightforward version of our attack, the attacker optimizes the perturbations using fixed model parameters for a number of steps (usually 250). However, this may lead to perturbations overfitting to a clean-trained model; during a real attack, a model is trained on poisoned data, but we optimize the poisons on a model that is trained only with clean data. To close the gap, we introduce model retraining during the poison crafting procedure. After retraining our model on the perturbed data, we again take optimization steps on the perturbations, but this time evaluating the training and adversarial losses at the new parameter vector. We repeat this process of retraining/optimizing several times and find that this noticeably improves the success of the poisons - often boosting success by more than 20% (see Tables 3, 10, 11).

See Appendix A.1 for an empirical evaluation of the importance of poison selection and model retraining for estimating the adversarial gradients of a victim. A brief description of our threat model is found in Algorithm 1.

---

**Algorithm 1** Sleeper Agent poison crafting procedure

---

**Input:** Training data $\mathcal{T} = \{(x_i, y_i)\}_{i=1}^{N}$, trigger patch $p$, source label $y_s$, target label $y_t$, poison budget $M \le N$, optimization steps $R$, retraining factor $T$

**Begin:**
1: Train surrogate network or ensemble $F(.\,; \theta)$ on training data $\mathcal{T}$
2: Select $M$ samples with label $y_t$ from $\mathcal{T}$ with highest gradient norm
3: Randomly initialize perturbations $\delta_{i=1}^{M}$
4: **for** $r = 1, 2, ..., R$ optimizations steps **do**
5:     Compute $\mathcal{A}(\delta, \theta, p, y_t, y_s)$ and update $\delta_{i=1}^{M}$ with a step of signed Adam
6:     **if** $r \mod \lfloor R/(T+1) \rfloor = 0$ and $r \ne R$ **then**
7:         Retrain $F$ on poisoned training data $\{(x_i + \delta_i, y_i)\}_{i=1}^{M} \cup \{(x_i, y_i)\}_{i=M+1}^{N}$ and update $\theta$
8:     **end if**
9: **end for**
10: **return:** poison perturbations $\delta_{i=1}^{M}$

---

Table 1: **Baseline evaluations** on CIFAR-10. Perturbations have $\ell_\infty$-norm bounded above by $16/255$, and poison budget is $1\%$ of training images.

| Architecture | ResNet-18 | MobileNetV2 | VGG11 |
|---|---|---|---|
| Clean model val (%) | 92.31 ($\pm 0.08$) | 88.19 ($\pm 0.05$) | 89.00 ($\pm 0.03$) |
| Poisoned model val (%) | 92.16 ($\pm 0.05$) | 88.03 ($\pm 0.05$) | 88.70 ($\pm 0.04$) |
| Clean model source val (%) | 92.36 ($\pm 0.93$) | 88.55 ($\pm 1.64$) | 90.62 ($\pm 1.23$) |
| Poisoned model source val (%) | 91.50 ($\pm 0.88$) | 87.79 ($\pm 1.60$) | 89.45 ($\pm 1.19$) |
| Poisoned model patched source val (%) | **12.96** ($\pm 5.40$) | **21.09** ($\pm 5.41$) | **17.97** ($\pm 4.00$) |
| Attack Success Rate (%) | **85.27** ($\pm 5.90$) | **72.92** ($\pm 6.09$) | **75.15** ($\pm 5.40$) |

Table 2: **The effect of poison budget.** Experiments on CIFAR-10 with ResNet-18 models [He et al., 2016]. Perturbations have $\ell_\infty$-norm $\leq 16/255$.

| Poison Budget | 50 (0.1%) | 100 (0.2%) | 250 (0.5%) | 400 (0.6%) | 500 (1%) |
|---|---|---|---|---|---|
| Clean model val (%) | 92.34 ($\pm 0.05$) | 92.36 ($\pm 0.04$) | 92.31 ($\pm 0.04$) | 92.15 ($\pm 0.08$) | 92.31 ($\pm 0.08$) |
| Poisoned model val (%) | 92.33 ($\pm 0.04$) | 92.34 ($\pm 0.05$) | 92.25 ($\pm 0.04$) | 92.12 ($\pm 0.06$) | 92.16 ($\pm 0.05$) |
| Clean model source val (%) | 93.01 ($\pm 0.69$) | 91.08 ($\pm 0.85$) | 92.43 ($\pm 0.74$) | 92.42 ($\pm 0.80$) | 92.36 ($\pm 0.93$) |
| Poisoned model source val (%) | 93.03 ($\pm 0.67$) | 90.61 ($\pm 0.86$) | 91.83 ($\pm 0.75$) | 91.88 ($\pm 0.79$) | 91.50 ($\pm 0.88$) |
| Poisoned model patched source val (%) | **61.04** ($\pm 4.27$) | **40.07** ($\pm 5.72$) | **22.77** ($\pm 4.77$) | **15.88** ($\pm 4.91$) | **12.96** ($\pm 5.40$) |
| Attack Success Rate (%) | **24.71** ($\pm 4.10$) | **49.76** ($\pm 6.21$) | **72.48** ($\pm 5.24$) | **81.44** ($\pm 5.25$) | **85.27** ($\pm 5.90$) |

## 4 Experiments

In this section, we empirically test the proposed Sleeper Agent backdoor attack on multiple datasets, against black-box settings, using an existing benchmark, and against popular defenses. Details regarding the experimental setup can be found in Appendix B.

### 4.1 Baseline Evaluations

Typically, backdoor attacks are considered successful if poisoned models do not suffer from a significant drop in validation accuracy on images without triggers, but they reliably misclassify images from the source class into the target class when a trigger is applied. We begin by testing our method in the gray-box setting. In the gray-box setting, we use the same architecture but different random initialization for crafting poisons and testing. Table 1 depicts the performance of Sleeper Agent on CIFAR-10 when perturbing $1\%$ of images in the training set with each perturbation constrained in an $\ell_\infty$-norm ball of radius $16/255$. During poison crafting, the surrogate model undergoes four evenly spaced retraining periods ($T = 4$), and we test the effectiveness of each surrogate model architecture at generating poisons for victim models of the same architecture. In subsequent sections, we will extend these experiments to the black-box setting and to an ensemblized attacker. We observe in these experiments that the poisoned models indeed achieve very similar validation accuracy to their clean counterparts, yet the application of triggers to source class images causes them to be misclassified into the target class as desired. In Table 2, we observe that Sleeper Agent can even be effective when the attacker is only able to poison a very small percentage of the training set. Note that the success of backdoor attacks depends greatly on the choice of source and target classes, especially since some classes contain very large objects which may dominate the image, even when a trigger is inserted. As a result, the variance of attack performance is high since we sample class pairs randomly. The poisoning and victim hyperparameters we use for our experiments can be found in Appendix B.

**The benefits of ensembling:** One simple way we can improve the transferability of our backdoor attack across initializations of the same architecture is to craft our poisons on an ensemble of multiple copies of the same architecture but trained using different initializations and different batch sampling during their training procedures. This behavior has also been observed in Huang et al. [2020b], Geiping et al. [2021]. In Table 3, we observe that this ensembling strategy indeed can offer significant performance boosts, both with and without retraining.

**The black-box setting:** Now that we have established the transferability of Sleeper Agent across models of the same architecture, we test on the hard black-box scenario where the victim's architecture is completely unknown to the attacker. This setting has proven extremely challenging for existing methods [Schwarzschild et al., 2021]. Table 4 contains four settings. In the first row, we simply craft the poisons on a single ResNet-18 and transfer these to other models. Second, we craft poisons

Table 3: **Ensembles** consisting of copies of the same architecture (ResNet-18). $S$ denotes the size of the ensemble, and $T$ denotes the retraining factor. Experiments are conducted on CIFAR-10, perturbations have $\ell_\infty$-norm bounded by $16/255$, and the attacker can poison $1\%$ of training images.

| Attack | Clean model val (%) | Poisoned model val (%) | Attack Success Rate (%) |
|---|---|---|---|
| Sleeper Agent ($S = 1, T = 0$) | 92.36 ($\pm 0.05$) | 92.08 ($\pm 0.08$) | 63.49 ($\pm 6.13$) |
| Sleeper Agent ($S = 2, T = 0$) | 92.10 ($\pm 0.04$) | 92.12 ($\pm 0.06$) | 64.70 ($\pm 5.65$) |
| Sleeper Agent ($S = 4, T = 0$) | 92.14 ($\pm 0.03$) | 91.98($\pm 0.05$) | **74.81** ($\pm 4.10$) |
| Sleeper Agent ($S = 2, T = 4$) | 92.11 ($\pm 0.07$) | 92.08 ($\pm 0.13$) | 87.40 ($\pm 6.23$) |
| Sleeper Agent ($S = 4, T = 4$) | 92.17 ($\pm 0.03$) | 91.81 ($\pm 0.06$) | **88.45** ($\pm 6.00$) |

Table 4: **Black-box attacks:** First row: Attacks crafted on a single ResNet-18 and transferred. Second row: attacks crafted on MobileNet-V2 and ResNet-34 and transferred. Third row: attacks crafted on the remaining architectures excluding the victim. The ensemble used in the last row includes the victim architecture. Experiments are conducted on CIFAR-10 and perturbations have $\ell_\infty$-norm bounded above by $16/255$, and the attacker can poison $1\%$ of training images.

| Attack | ResNet-18 | MobileNet-V2 | VGG11 | Average |
|---|---|---|---|---|
| Sleeper Agent ($S = 1, T = 4$, ResNet-18) | – | 29.10% | 31.96% | 29.86% |
| Sleeper Agent ($S = 4, T = 0$, MobileNet-V2, ResNet-34) | 70.30% | – | 46.48% | 58.44% |
| Sleeper Agent ($S = 4, T = 0$, victim excluded) | 63.11% | 42.40% | 55.28% | 53.60% |
| Sleeper Agent ($S = 6, T = 0$, victim included) | 68.46% | 67.28% | 85.37% | 73.30% |

on an ensemble consisting of two MobileNet-V2 and two ResNet-34 architectures and transfer to the remaining models. Third, for each architecture, we craft poisons with an ensemble consisting of the other two architectures and test on the remaining one. The second and third scenarios are ensemblized black-box attacks, and we see that Sleeper Agent is effective. In the last row, we perform the same experiment but with the testing model included in the ensemble, and we observe that a single ensemble can craft poisons that are extremely effective on a range of architectures. We choose ResNet-18, MobileNet-V2, and VGG11 as these are common and contain a wide array of structural diversity [He et al., 2016, Sandler et al., 2018, Simonyan and Zisserman, 2014]. Additionally, Guo and Liu [2020] considers the case that the attacker uses a weaker surrogate than the defender's model. We simulate this case by using a VGG11 surrogate and ResNet-18 target. We find, with a $1\%$ poison budget on CIFAR-10, that Sleeper Agent achieves an attack success rate of $57.47\%$.

**ImageNet evaluations:** In addition to CIFAR-10, we perform experiments on ImageNet. Table 5 summarizes the performance of Sleeper Agent on ImageNet where attacks are crafted and tested on ResNet-18 and MobileNetV2 models. Each attacker can only perturb $0.05\%$ of training images, and perturbations are constrained in an $\ell_\infty$-norm ball of radius $16/255$ - a bound seen in prior poisoning works on ImageNet [Fowl et al., 2021a, Geiping et al., 2021, Saha et al., 2020]. To have a strong threat model, we use the retraining factor of two ($T = 2$) so that the surrogate model is retrained at two evenly spaced intervals. Figure 1b contains visualizations of the patched sources and the crafted poisons. The details of models and hyperparameters can be found in Appendix B. Additional experiments on ImageNet and further visualizations are presented in Appendices A and C.

## 4.2 Comparison to Other Methods

There are several existing clean-label hidden-trigger backdoor attacks that claim success in settings different than ours. In order to further demonstrate the success of our method, we compare our poisons to ones generated from these methods in our strict threat model of from-scratch training. In these experiments, poisons are generated by our attack, clean label backdoor, and hidden trigger backdoor. All poison trials have the same randomly selected source-target class pairs, the same

Table 5: **ImageNet evaluations**. Perturbations have $\ell_\infty$-norm bounded above by $16/255$, and the poison budget is $0.05\%$ of training images.

| Architecture | ResNet-18 | MobileNetV2 |
|---|---|---|
| Clean model val (%) | 69.76 | 71.88 |
| Poisoned model val (%) | 67.84 ($\pm 0.10$) | 68.60 ($\pm 0.03$) |
| Attack Success Rate (%) | **44.00** ($\pm 6.73$) | **41.00** ($\pm 3.31$) |

Table 6: **Benchmark results on CIFAR-10**. Comparison of our method to popular "clean-label" attacks. Results averaged over the same source/target pairs with $\epsilon = 16/255$ and poison budget $1\%$.

| Attack | ResNet-18 | MobileNetV2 | VGG11 | Average |
|---|---|---|---|---|
| Hidden-Trigger Backdoor [Saha et al., 2020] | 3.50% | 3.76% | 5.02% | 4.09% |
| Clean-Label Backdoor [Turner et al., 2019] | 2.78% | 3.50% | 4.70% | 3.66% |
| Sleeper Agent (Ours) | **78.84**% | **75.96**% | **86.60**% | **80.47**% |

budget, and the same $\varepsilon$-bound (Note: clean-label backdoor originally did not use $\ell_\infty$ bounds, so we adjust the opacity of their perturbations to ensure the constraint is satisfied). We then train a randomly initialized network from scratch on these poisons and evaluate success over 1000 patched source images. We test three popular architectures and find that our attack significantly outperforms both methods and is the only backdoor method to exceed single digit success rates, confirming the findings of Schwarzschild et al. [2021] on the fragility of these existing methods. See Table 6 for full results.

## 4.3 Defenses

A selling point for hidden trigger backdoor attacks is that the trigger that is used to induce misclassification at test-time is not present in any training data, thus making inspection based defenses, or automated pattern matching more difficult. However, there exist numerous defenses, aside from visual inspection, that have been proposed to mitigate the effects of poisoning - both backdoor and other attacks. We test our method against a number of popular defenses.

**Spectral Signatures**: This defense, proposed in Tran et al. [2018], aims to filter a pre-selected amount of training data based upon correlations with singular vectors of the feature covariance matrix. This defense was originally intended to detect triggers used in backdoor attacks.

**Activation Clustering**: Chen et al. [2019] clusters activation patterns to detect anomalous inputs. Unlike the spectral signatures defense, this defense does not filter a pre-selected volume of data.

**DPSGD**: Poison defenses based on differentially private SGD [Abadi et al., 2016] have also been proposed [Hong et al., 2020]. Differentially private learning inures models to small changes in training data, which provably imbues robustness to poisoned data.

**Data Augmentations**: Recent work has suggested that strong data augmentations, such as mixup, break data poisoning [Borgnia et al., 2021]. This has been confirmed in recent benchmark tests which demonstrate many poisoning techniques are brittle to slight changes in victim training routine [Schwarzschild et al., 2021]. We test against mixup augmentation [Zhang et al., 2018].

**STRIP**: Gao et al. [2019] proposes to add strong perturbations by superimposing input images at test time to detect the backdoored inputs based on the entropy of the predicted class distribution. If the entropy is lower than a predefined threshold, the input is considered backdoored and is rejected.

**NeuralCleanse**: Wang et al. [2019] proposes a defense designed for traditional backdoor attacks by reconstructing the maximally adversarial trigger used to backdoor a model. While this defense was not designed for hidden trigger backdoor attacks, we experiment with this as a *detection* defense wherein we test whether NeuralCleanse can detect the backdoored class. This modification is denoted by NeuralCleanse*. In our trials, NeuralCleanse* does not successfully detect any of the backdoored classes - as determined by taking the maximum mask MAD (see Wang et al. [2019]). Neural Cleanse does not produce an anomaly score $> 2$ (their characterization of detecting outliers) for the backdoored class in *any* of our experiments.

We find that across the board, all of these defenses exhibit a robustness-accuracy trade-off. Many of these defenses do not reliably nullify the attack, and defenses that do degrade attack success also induce such a large drop in validation accuracy that they are unattractive options for practitioners. For example, to lower the attack success to an average of $13.14\%$, training with DPSGD degrades natural accuracy on CIFAR-10 to $70\%$. See Table 7 for the complete results of these experiments. Additional evaluations on recent defenses are presented in Appendix A.6.

Table 7: **Defenses**. Experiments are conducted on CIFAR-10 with ResNet-18 models, perturbations have $\ell_\infty$-norm bounded above by $16/255$, and poison budget is $1\%$ of training images.

| Defense | Attack Success Rate (%) | Validation Accuracy (%) |
|---|---|---|
| Spectral Signatures | 37.17 ($\pm 10.10$) | 89.94 ($\pm 0.19$) |
| Activation Clustering | 15.17 ($\pm 5.38$) | 72.38 ($\pm 0.48$) |
| DPSGD | 13.14 ($\pm 4.49$) | 70.00 ($\pm 0.17$) |
| Data Augmentation | 69.75 ($\pm 10.77$) | 91.32 ($\pm 0.12$) |
| STRIP | 62.68 ($\pm 4.90$) | 92.23 ($\pm 0.05$) |
| NeuralCleanse* | 85.27 ($\pm 5.90$) | 92.31 ($\pm 0.08$) |

Table 8: **Random poisons**. Experiments are conducted on CIFAR-10 with ResNet-18 models. Perturbations have $\ell_\infty$-norm bounded above by $16/255$ and poisons are drawn from all classes.

| Attack | Poison budget | Attack Success Rate (%) |
|---|---|---|
| Sleeper Agent (S = 1, T = 4) | 1% | **41.90** ($\pm 7.16$) |
| Sleeper Agent (S = 1, T = 4) | 3% | **66.51** ($\pm 6.90$) |

## 4.4 Sleeper Agent Can Poison Images in Any Class

Typical backdoor attacks which rely on label flips or feature collisions can only function when poisons come from the source and/or target classes [Saha et al., 2020, Turner et al., 2019]. This restriction may be a serious limitation in practice. In contrast, we show that Sleeper Agent can be effective even when we poison images drawn from all classes. To take advantage of our data selection strategy, we select poisons with maximum gradient norm across all classes. Table 8 contains the performance of Sleeper Agent in the aforementioned setting.

## 4.5 Evaluations Under Hard $\ell_\infty$-norm Constraints

While existing works on backdoor attacks consider poisons with $\ell_\infty$-norm bounded above by $16/255$ as an imperceptible threat [Saha et al., 2020, Turner et al., 2019], Nguyen and Tran [2020] shows that human inspection can detect poisoned samples effectively. This inspection might mitigate the threat of large perturbations. To bypass this possibility, we conduct our baseline experiments on CIFAR-10 using perturbations with small $\ell_\infty$-norms. From Table 9, we observe that our threat model is effective even with an $\ell_\infty$-norm bounded above by $8/255$. Visualizations can be found in Appendix C.

## 4.6 Ablation Studies

Here, we analyze the importance of each technique in our algorithm via ablation studies. We focus on three aspects of our method: 1) patch location, 2) retraining during poison crafting, 3) poison selection, and 4) retraining factor. Table 10 details the combinations and their effects on poison success. We find that randomizing patch location improves poisoning success, and both retraining and data selection based on maximum gradient significantly improve poison performance. Combining all three boosts poison success more than four-fold. To further show the importance of retraining, we conduct more experiments with and without retraining on ImageNet. From Table 11, we infer that retraining is essential. Additional ablations studies are found in Appendix A.

Table 9: **Evaluation under different $\ell_\infty$-norm**. Experiments are conducted on CIFAR-10 with ResNet-18 models, and the poison budget is $1\%$ of training images.

| Perturbation $\ell_\infty$-norm | Attack Success Rate (%) |
|---|---|
| 8/255 | 37.32 ($\pm 8.33$) |
| 10/255 | 55.75 ($\pm 8.12$) |
| 12/255 | 63.31 ($\pm 8.84$) |
| 14/255 | 78.03 ($\pm 7.13$) |
| 16/255 | 85.27 ($\pm 5.90$) |

Table 10: **CIFAR-10 ablation studies.** Investigation of the effects of random patch-location, retraining, and data selection. Experiments are conducted on CIFAR-10 with ResNet-18 models, perturbations have $\ell_\infty$-norm bounded above by $16/255$, and poison budget is $1\%$ of training images.

| Attack setup | Attack Success Rate (%) |
|---|---|
| Fix patch-location (bottom-right corner) | 19.25 ($\pm$3.01) |
| Random patch-location | 33.95 ($\pm$4.57) |
| Random patch-location + retraining | 59.42 ($\pm$5.78) |
| Random patch-location + data selection | 63.49 ($\pm$6.13) |
| Random patch-location + retraining + data selection | **85.27** ($\pm$5.90) |

Table 11: **ImageNet ablation studies**. Perturbations have $\ell_\infty$-norm bounded above by $16/255$, and the poison budget is $0.05\%$ of training images.

| Attack | Attack Success Rate (%) |
|---|---|
| Sleeper Agent (S = 1, T = 0) | 22.00 ($\pm$5.65) |
| Sleeper Agent (S = 1, T = 2) | **44.00** ($\pm$6.73) |

## 5 Broader Impact and Limitations

In this work, we illuminate a new scalable backdoor attack that could be used to stealthily compromise security-critical systems. We hope that by highlighting the potential danger of this nefarious threat model, our work will give rise to stronger defenses and will encourage caution on the part of practitioners.

While on average, our method is effective, the variance is large, and the success of our method can range from almost all patched images being misclassified to low success. This behavior has previously been observed in Schwarzschild et al. [2021]. In real-world scenarios, datasets are often noisy and imbalanced, so training behavior may be mysterious. As a result, practitioners should be cautious in their expectations that methods developed on datasets like CIFAR-10 and ImageNet will work on their own problems.

## 6 Conclusion

In this work, we present the first hidden-trigger backdoor attack that is effective against deep networks trained from scratch. This is a challenging setting for backdoor attacks, and existing attacks typically operate in less strict settings. Nonetheless, we choose the strict setting because practitioners often train networks from scratch in real-world applications, and patched poisons may be easily visible upon human inspection. In order to accomplish the above goal, we use a gradient matching objective as a surrogate for the bilevel optimization problem, and we add features such as re-training and data selection in order to significantly enhance the performance of our method, Sleeper Agent.

## Acknowledgements

This work was supported by DARPA GARD under contracts #HR00112020007 and HR001119S0026-GARD-FP-052, the DARPA YFA program, the ONR MURI Program under the Grant N00014-20-1-2787, and the National Science Foundation DMS program. Further support was provided by JP Morgan Chase and Capital One Bank.

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
