# A Additional Experiments

## A.1 Gradient Alignment Throughout Training

In order to demonstrate that the gradients of the poison examples are well aligned with the adversarial gradient throughout the training of the victim model, we visualize the cosine similarity between the adversarial gradient and the poison examples in multiple settings across epochs of training. Figure 2 contains three experiments. First, we train a clean model where the attack's success rate is very low (almost zero). Second, we train a poisoned model without data selection or retraining. And third, we employ poisons that have been generated utilizing data selection and retraining techniques. As shown in Table 10, the average attack success rate for the second and third experiments is $33.95\%$ and $85.27\%$, respectively. Figure 2 shows that a successful attack yields far superior gradient alignment and hence a high attack success rate. In addition, these experiments demonstrate that gradient alignment, data selection, and retraining all work together collaboratively.

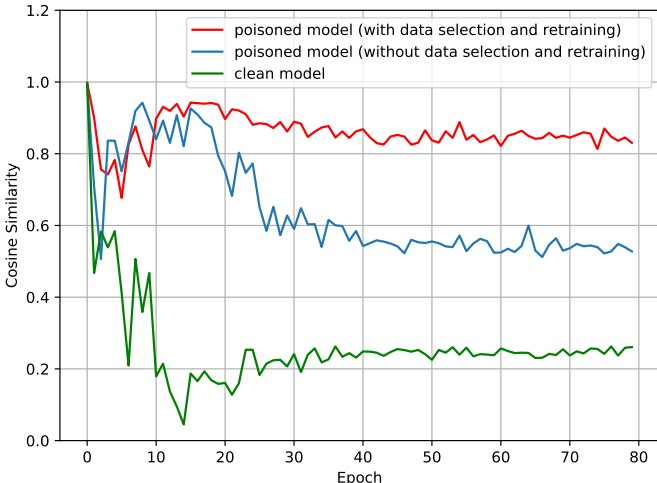

Figure 2: Cosine Similarity, per epoch, between the adversarial gradient $\nabla_\theta \mathcal{L}_{adv}$ and gradient of the poison examples (clean examples from the target class in the case of clean model training) $\nabla_\theta \mathcal{L}_{train}$ for two different poisoned models and a clean model. Experiments are conducted on CIFAR-10 with ResNet-18 models.

## A.2 Patch Choice

Sleeper Agent is designed in a way that the backdoor attack is efficient for any random patch the threat model uses for crafting poisons. To show this, we conduct the same baseline experiments discussed in Section 4.1 using different random patches that are generated using a Bernoulli distribution. From Table 12, we observe that the choice of the patch does not affect Sleeper Agent's success rate. Figure 3 depicts few samples of the random patches we use for the experiments presented in Table 12.

Table 12: **Baseline evaluations using random patches** on CIFAR-10. Perturbations have $\ell_\infty$-norm bounded above by $16/255$, and poison budget is $1\%$ of training images. Each number denotes an average (and standard error) over $24$ independent crafting and training runs along with randomly sampled source/target class pairs. Each run has a unique patch generated randomly.

| Architecture | ResNet-18 |
|---|---|
| Clean model val(%) | 92.16 ($\pm$0.08) |
| Poisoned model val (%) | 92.00 ($\pm$0.07) |
| Clean model source val (%) | 92.55 ($\pm$0.98) |
| Poisoned model source val (%) | 91.77 ($\pm$1.09) |
| Poisoned model patched source val (%) | **14.86** ($\pm$5.06) |
| Attack Success Rate (%) | **82.05** ($\pm$5.80) |

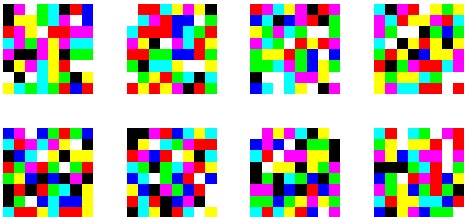

Figure 3: Sample random patches

## A.3 Patch Size

To investigate the effect of patch size on the attack success rate, we perform the baseline evaluation discussed in Section 4.1 using different patch sizes. From Table 13, we observe that by poisoning only $0.05\%$ of the training set and using a larger patch, we can effectively poison ImageNet. Furthermore, by using a proper amount of perturbation, Sleeper Agent works well with the smaller patches on both CIFAR-10 and ImageNet datasets. Visualizations of patched sources using different patch sizes are shown in Figure 9.

Table 13: **The effect of patch size**. Experiments are conducted on CIFAR-10 and ImageNet datasets with ResNet-18 models. Visualizations of different patched sources from ImageNet dataset can be found in Figure 9.

| Attack | Dataset | Poison budget | Patch size | $\ell_\infty$-norm | Attack Success Rate (%) |
|---|---|---|---|---|---|
| Sleeper Agent (S = 1, T = 4) | CIFAR-10 | $1\%$ | $6 \times 6$ | $20/255$ | 64.78 |
| Sleeper Agent (S = 1, T = 4) | CIFAR-10 | $1\%$ | $8 \times 8$ | $16/255$ | 85.27 |
| Sleeper Agent (S = 1, T = 2) | ImageNet | $0.05\%$ | $25 \times 25$ | $16/255$ | 38.00 |
| Sleeper Agent (S = 1, T = 2) | ImageNet | $0.05\%$ | $25 \times 25$ | $24/255$ | 52.00 |
| Sleeper Agent (S = 1, T = 2) | ImageNet | $0.05\%$ | $30 \times 30$ | $16/255$ | 44.00 |
| Sleeper Agent (S = 1, T = 2) | ImageNet | $0.05\%$ | $45 \times 45$ | $16/255$ | 50.50 |

## A.4 More Evaluations on ImageNet

In addition to the experiments in Section 4.1 and Appendix A.3, we provide more evaluations on ImageNet dataset focusing on low poison budget and smaller $\ell_\infty$-norm constraint. The evaluation results are listed in Table 14. The results indicate that our proposed threat model is still effective by poisoning only 250 images in the ImageNet trainset. Additionally, under the hard $\ell_\infty$-norm constraint of $8/255$, Sleeper Agent has a partial success of one out of four (significantly better than random guess with a success rate of $0.001$ on ImageNet).

Table 14: **ImageNet evaluations**. Experiments are conducted on ResNet-18 models.

| Attack | Perturbation $\ell_\infty$-norm | Poison budget | Attack Success Rate (%) |
|---|---|---|---|
| Sleeper Agent (S = 1, T = 2) | **8/255** | $0.05\%$ (500 images) | 28.00 |
| Sleeper Agent (S = 1, T = 2) | $16/255$ | $0.025\%$ (**250 images**) | 27.33 |

## A.5 Retraining Factor

Table 15 demonstrates the effect of the retraining factor on the attack success rate on the CIFAR-10 dataset. For $T$ larger than 4, we do not see a considerable improvement in the attack success rate. Since increasing $T$ is costly, we choose $T = 4$ as it simultaneously gives us a high success rate and is also significantly faster than $T = 8$. We observe that even with $T = 4$, the attack success rate is above $95\%$ in most trials

Table 15: **Ablation studies on retraining factor**. Investigation of the effects of retraining factor $T$. Experiments are conducted on CIFAR-10 with ResNet-18 models, perturbations have $\ell_\infty$-norm bounded above by $16/255$, and the poison budget is $1\%$ of training images.

| Retraining factor | Attack Success Rate (%) |
|---|---|
| $T = 0$ | 63.49 ($\pm 6.13$) |
| $T = 2$ | 70.66 ($\pm 6.66$) |
| $T = 4$ | 85.27 ($\pm 5.90$) |
| $T = 8$ | 86.48 ($\pm 6.26$) |

## A.6 Additional Defenses

In addition to the defenses evaluated in Section 4.3, we evaluate Sleeper Agent on two recent defenses, ABL [Li et al., 2021a] and ANP [Wu and Wang, 2021]. We evaluate against ANP with various threshold values, and we report the accuracy of a ResNet-18 on CIFAR-10, where Sleeper Agent poisons have $\ell_\infty$-norm bounded above by $16/255$. We find that ANP cannot achieve a low attack success rate without greatly reducing the model's validation accuracy on clean data. Similarly, we find that ABL must decrease accuracy substantially to remove the backdoor vulnerability. Even after 20 epochs of unlearning, Sleeper Agent has nearly a 60% success rate. See Tables 16 and 17 for the complete results of these experiments.

Table 16: **Adversarial Neuron Pruning (ANP)**. We test Sleeper Agent against the defense, ANP [Wu and Wang, 2021], across various threshold levels. We see here that high threshold values can decrease the attack success rate but at a high cost in validation accuracy. Experiments are conducted on CIFAR-10 with ResNet-18 models. Perturbations have $\ell_\infty$-norm bounded above by $16/255$.

| Defense | Threshold | Validation Accuracy (%) | Attack Success Rate (%) |
|---|---|---|---|
| None | - | 92.31 | 85.27 |
| ANP | 0.05 | 80.05 | 51.03 |
| ANP | 0.10 | 71.75 | 27.87 |
| ANP | 0.15 | 50.47 | 17.77 |
| ANP | 0.20 | 16.56 | 3.35 |

Table 17: **Anti-Backdoor Learning (ABL)**. We test Sleeper Agent against the defense, ABL [Li et al., 2021a], across various numbers of unlearning epochs. We see here that many unlearning epochs can decrease the attack success rate but at a high cost in validation accuracy. Experiments are conducted on CIFAR-10 with ResNet-18 models. Perturbations have $\ell_\infty$-norm bounded above by $16/255$.

| Defense | Unlearning Epochs | Validation Accuracy (%) | Attack Success Rate (%) |
|---|---|---|---|
| None | - | 92.31 | 85.27 |
| ABL | 5 | 87.53 | 70.72 |
| ABL | 10 | 86.85 | 68.35 |
| ABL | 15 | 82.34 | 64.11 |
| ABL | 20 | 64.55 | 59.51 |

Table 18: **Adversarial Training**. We test Sleeper Agent against adversarial training via PGD with a perturbation radius of $4/255$. Experiments are conducted on CIFAR-10 with ResNet-18 models. Poison perturbations have $\ell_\infty$-norm bounded above by $16/255$.

| Defense | Validation Accuracy (%) | Attack Success Rate (%) |
|---|---|---|
| None | 92.31 | 85.27 |
| Adv. Training | 88.63 | 52.83 |

# B    Implementation Details

## B.1    Experimental Setup

The most challenging setting for evaluating a backdoor attack targets victim models that are trained from scratch [Schwarzschild et al., 2021]. On the other hand, it is crucial to compute the average attack success rate on all patched source images in the validation set to evaluate effectiveness reliably. Hence, to evaluate our backdoor attack, we first poison a training set using a surrogate model as described in Algorithm 1, then the victim model is trained in a standard fashion on the poisoned training set from scratch with random initialization. After the victim model is trained, to compute the *attack success rate*, we measure the average rate at which patched source images are successfully classified as the target class. To be consistent and to provide a fair comparison to Saha et al. [2020], in our primary experiments, we use a random patch selected from Saha et al. [2020] as shown in Figure 4. In our baseline experiments, following Saha et al. [2020], the patch size is $8 \times 8$ for CIFAR-10 ($6.25\%$ of the pixels) and $30 \times 30$ for the ImageNet ($1.79\%$ of the pixels). Note that the choice of the patch in our implementation is not essential, and our model is effective across randomly selected patches (see Appendix A.2). More experiments on smaller patch sizes are presented in Appendix A.3.

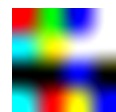

Figure 4: The trigger we use in our primary experiments.

## B.2    Models and Hyperparameters

For our evaluations, we use ResNet-18, ResNet-34, MobileNet-v2, and VGG11 [He et al., 2016, Sandler et al., 2018, Simonyan and Zisserman, 2014]. For training ResNet-18 and ResNet-34, we use initial learning rate $0.1$, and for MobileNet-v2 and VGG11, we use initial learning rate $0.01$. We schedule learning rate drops at epochs $14$, $24$, and $35$ by a factor of $0.1$. For all models, we employ SGD with Nesterov momentum, and we set the momentum coefficient to $0.9$. We use batches of $128$ images and weight decay with a coefficient of $4 \times 10^{-4}$. For all CIFAR-10 experiments, we train and retrain for $40$ epochs, and for validation, we train the re-initialized model for $80$ epochs. For the ImageNet experiments, we employ pre-trained models from `torchvision` to start crafting, and for retraining and validation, we apply a similar procedure explained: training for $80$ epochs for both retraining and validation while we schedule learning rate drops at epochs $30$, $50$, and $70$ by a factor of $0.1$. To incorporate data augmentation, for CIFAR-10, we apply horizontal flips with probability $0.5$ and random crops of size $32 \times 32$ with zero-padding of $4$. And for the ImageNet, we use the following data augmentations: 1) resize to $256 \times 256$, 2) central crop of size $224 \times 224$, 3) horizontal flip with probability $0.5$, 4) random crops of size $224 \times 224$ with zero-padding of $28$.

## B.3    Implementation of Benchmark Experiments

In Section 4.2 we compared our threat model with Clean-Label Backdoor Turner et al. [2019] and Hidden-Trigger Backdoor Saha et al. [2020]. For both methods, We follow the same procedure used in their papers as described in Schwarzschild et al. [2021]. Specifically, to reproduce the clean-label attack, we use the implementation code provided in Schwarzschild et al. [2021]. To get each poison, we compute the PGD-based adversarial perturbation to each image, and then the trigger is added to the image [Schwarzschild et al., 2021, Turner et al., 2019].

## B.4    Defense Details

In Section 4.3, we test a variety of defenses against our proposed attack. For filtering-based defenses, such as SS, AC, we craft the poisons as usual (according to Algorithm 1). We then train a model on the poisoned data. After this, we apply one of the selected defenses to identify what training data may have been poisoned. We then remove the detected samples, and retrain a *second* network from scratch on the remaining data. Finally, we evaluate the attack success rate (on the backdoored class) using this second network. For STRIP, we simply apply the defense at test time for the first

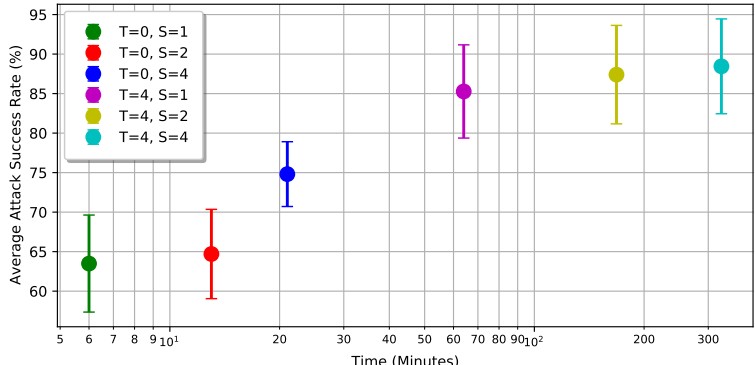

Figure 5: Average poisoning time for various Sleeper Agent setups. All experiments are conducted on CIFAR-10 with ResNet-18 models. Perturbations have $\ell_\infty$-norm bounded above by $16/255$, and the poison budget is $1\%$ of training images. $T$ denotes the training factor and $S$ denotes the ensemble size.

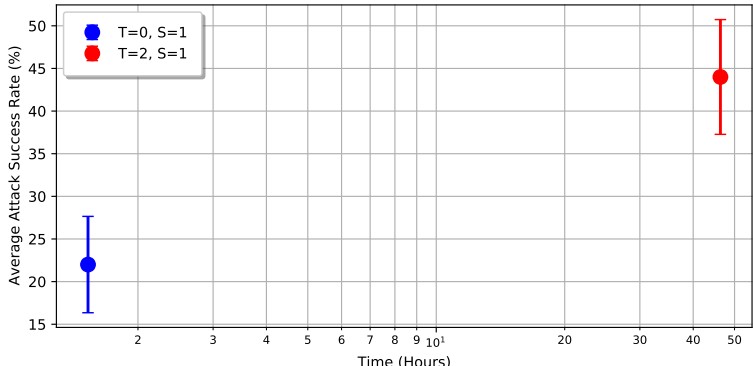

Figure 6: Average poisoning time for various Sleeper Agent setups. All experiments are conducted on ImageNet with ResNet-18 models. Perturbations have $\ell_\infty$-norm bounded above by $16/255$, and the poison budget is $0.05\%$ of training images. $T$ denotes the training factor and $S$ denotes the ensemble size.

network, and filter out any patched images that exceed an entropy threshold in their predictions. In this case, an attack is considered a success if a backdoored input is not detected at test time, *and* misclassified as the target class.

### B.5 Runtime Cost

We use two NVIDIA GEFORCE RTX 2080 Ti GPUs for baseline evaluations on CIFAR-10 and two to four NVIDIA GEFORCE RTX 3090 GPUs for ImageNet baseline evaluations depending on the network size. Figures 5 and 6 show the time cost of Sleeper Agent with different settings.

## C Visualizations

In this section, we present more visualizations of the successful attacks on CIFAR-10 and ImagNet datasets. Figures 7, 8, 9, 10, and 11 show patched sources and poisoned targets generated by Sleeper Agent on CIFAR-10 and ImageNet. We observe that the generated perturbed images and their corresponding clean images are hardly distinguishable by the human eye, especially in the last column of Figure 11 where the $\ell_\infty$-norm of perturbation is bounded above by $8/255$.

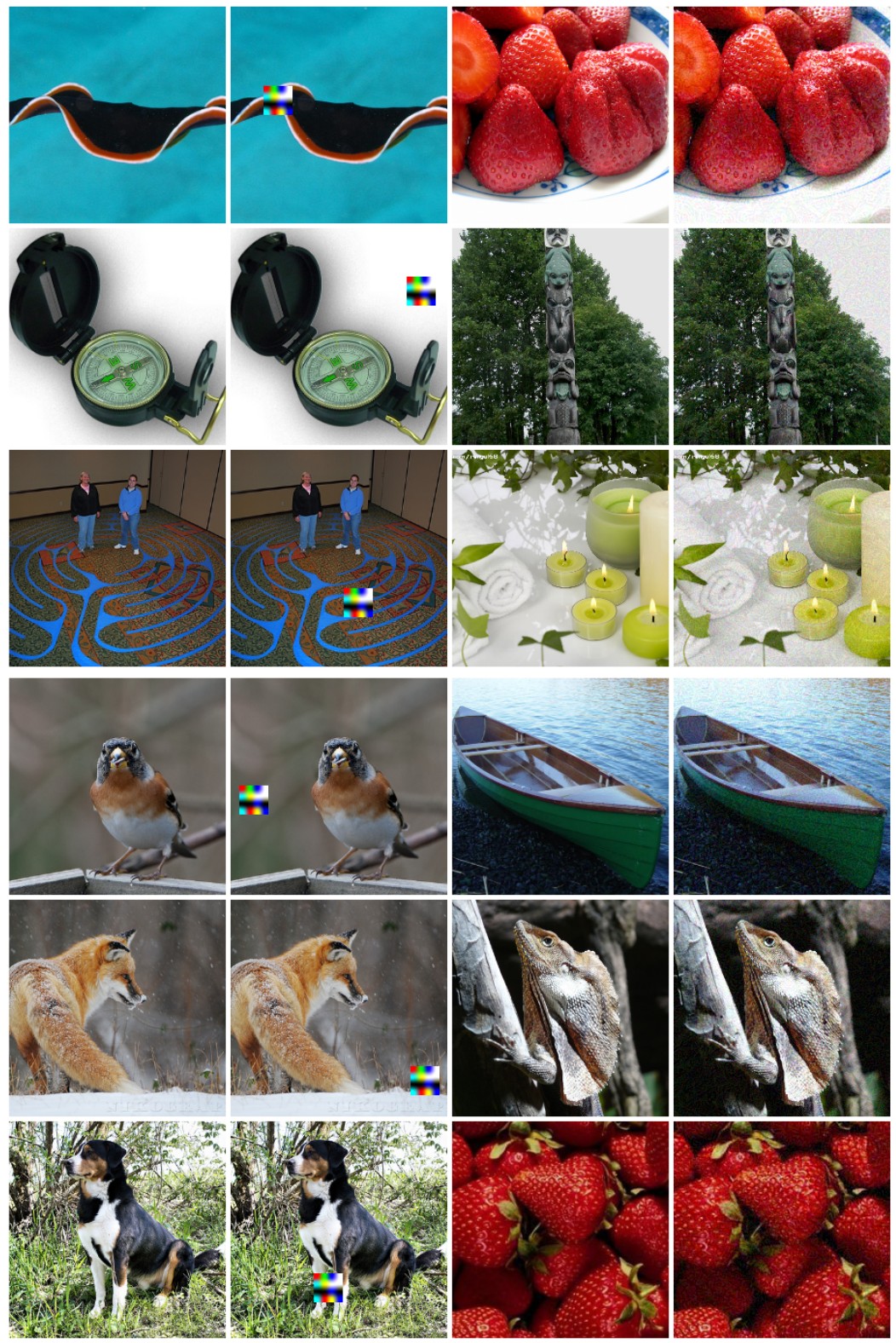

Figure 7: Visualizations of the successful attacks on the ImageNet dataset. Each row includes the clean source, patched source, clean target, and poisoned target, respectively. Perturbations have $\ell_\infty$-norm bounded above by $16/255$, and the patch size is $30$.

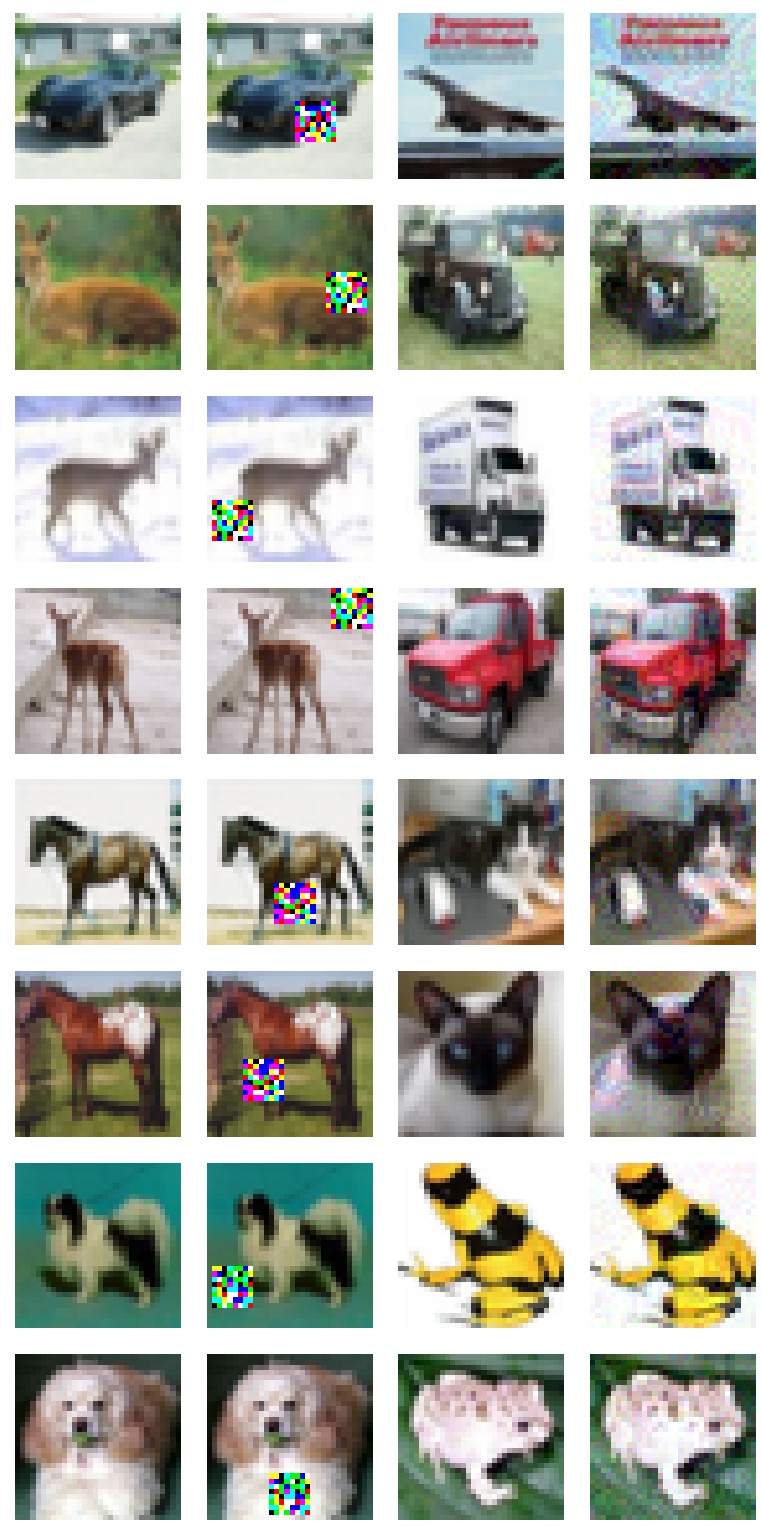

Figure 8: Visualizations of the successful attacks on the CIFAR-10 dataset. Each row includes the clean source, patched source, clean target, and poisoned target, respectively. Perturbations have $\ell_\infty$-norm bounded above by $16/255$ and the patch size is $8$. Here, patches are randomly generated as described in Appendix A.2.

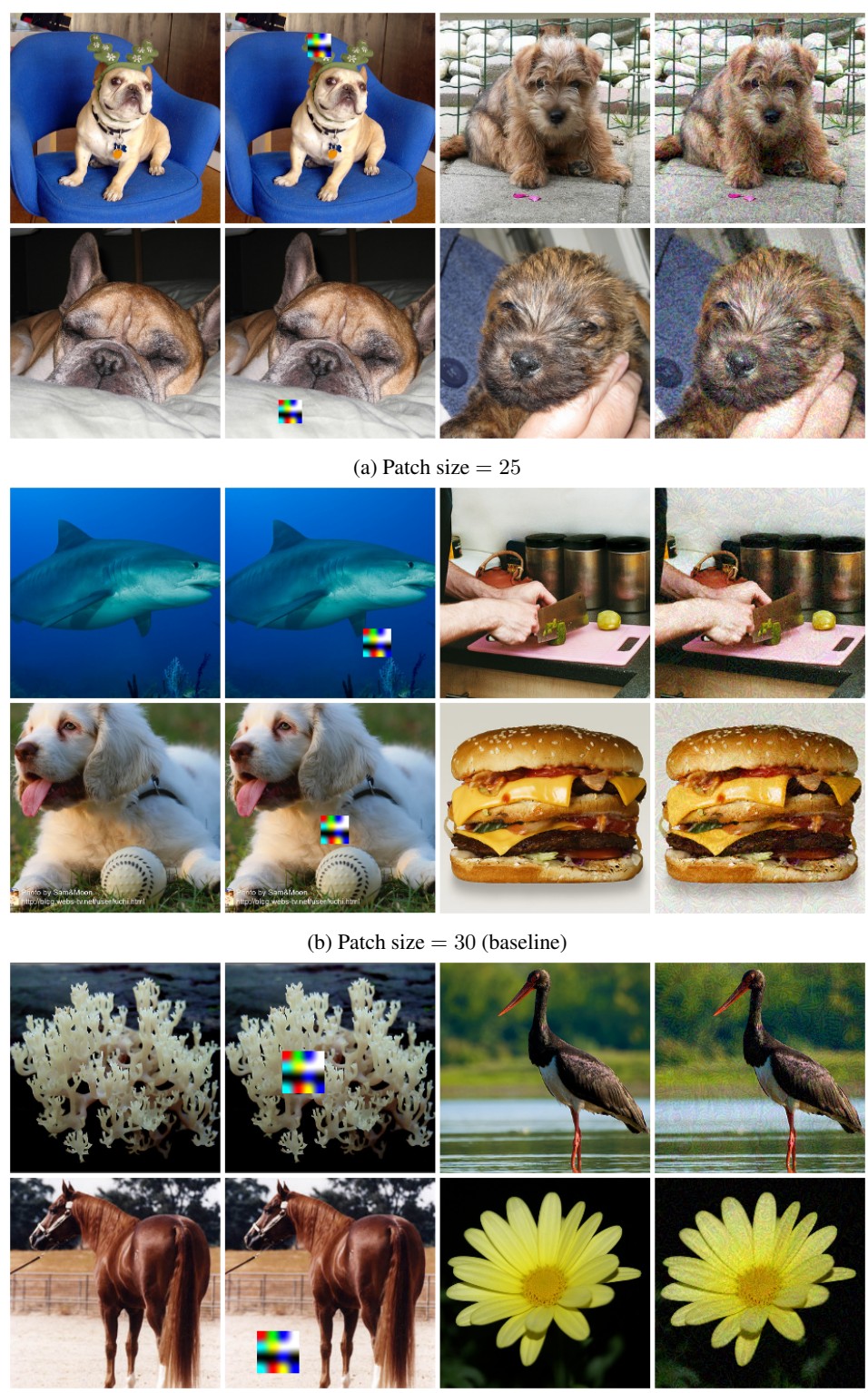

(a) Patch size = 25

(b) Patch size = 30 (baseline)

(c) Patch size = 45

Figure 9: Sample clean source (first column), patched source (second column), clean target (third column), and poisoned target (fourth column) from the ImageNet dataset with different trigger size. Perturbations have $\ell_\infty$-norm bounded above by $16/255$.

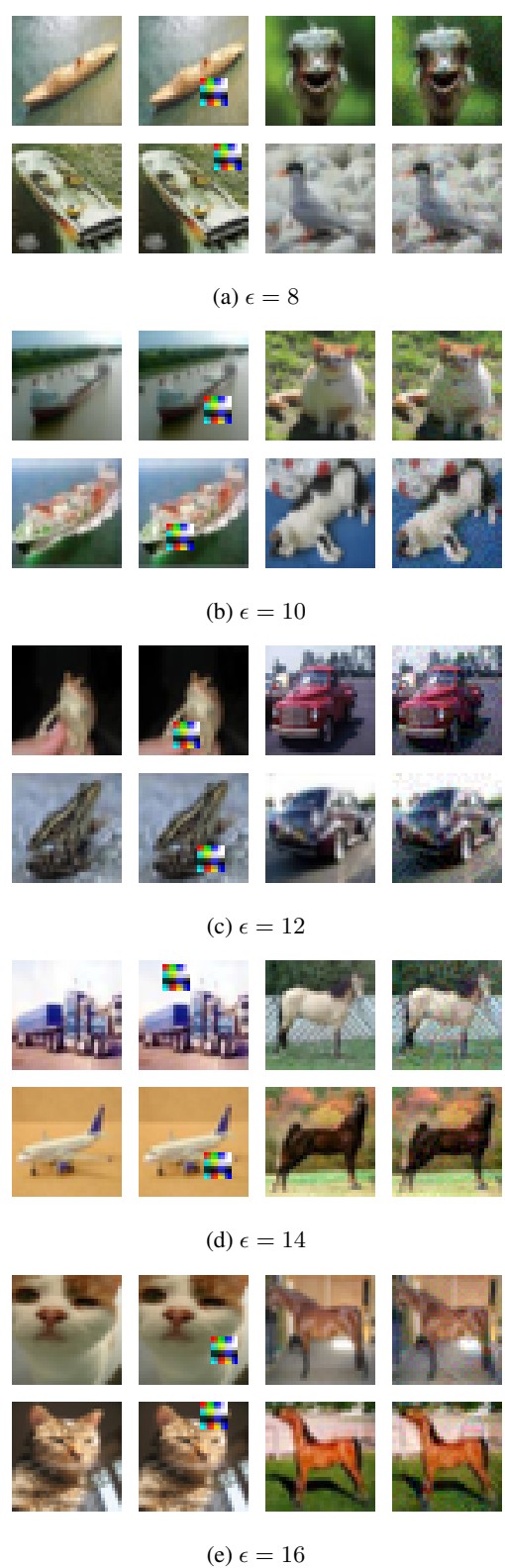

(a) $\epsilon = 8$

(b) $\epsilon = 10$

(c) $\epsilon = 12$

(d) $\epsilon = 14$

(e) $\epsilon = 16$

Figure 10: Sample clean source (first column), patched source (second column), clean target (third column), and poisoned target (fourth column) from the CIFAR-10 dataset with different $\ell_\infty$-norm perturbation.

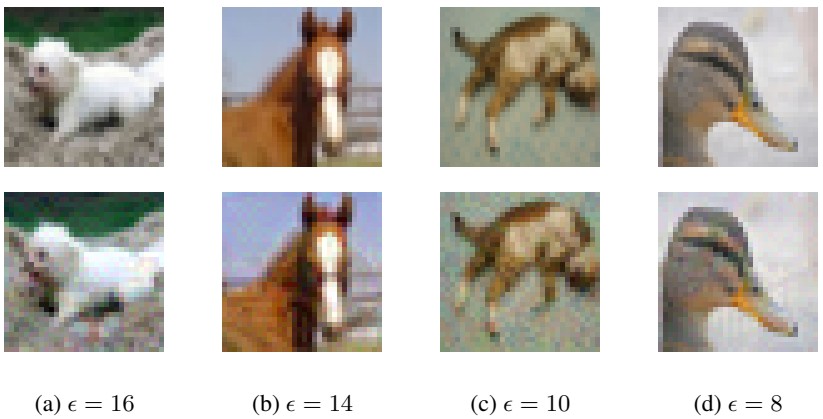

(a) $\epsilon = 16$        (b) $\epsilon = 14$        (c) $\epsilon = 10$        (d) $\epsilon = 8$

Figure 11: Visualization of clean targets (first row) and poisoned targets (second row) with different $\ell_\infty$-norms from the CIFAR-10 dataset.