# OpenReview forum: "Sleeper Agent: Scalable Hidden Trigger Backdoors for Neural Networks Trained from Scratch"
_NeurIPS.cc/2022/Conference — NeurIPS 2022 Accept_

### Official Review · Reviewer_vXHc · 2022-06-14

**Rating:** 5
**Confidence:** 4
**Soundness:** 2 fair
**Presentation:** 3 good
**Contribution:** 2 fair

**Summary:**

This paper presents a bi-level optimization based method for hidden trigger backdoor attacks (HTBAs), one type of backdoor attacks that the poisoned samples do not contain the exact trigger pattern (poisoning noise and the actual trigger are different). Three techniques are introduced for generating strong HTBAs, including 1) bi-level optimization with gradient matching, 2) poison selection, and 3) model retraining. The proposed attack is able to achieve high attack success rates for models trained from scratch, which is a substantial improvement over an existing fine-tuning based HTBA. Ablation studies are also conducted to help understand the importance of three techniques individually.

**Questions:**

1. The HTBA is not very common in the current literature and thus needs more clear and intuitive explanations, particularly in relation to the more extensively studied “invisible backdoor attacks”. As otherwise, Figure 1(b) is hard to understand what the columns are for. Fig. (a) also needs to include the attack step, to show the actual trigger used to mount the attack is different from the trigger patterns.
2. In Equation (1), the definition of $p$ (and how it relates to $\delta$), $\mathcal{D}$ (and its difference to  $\mathcal{T}$) should be clearly stated. It is not so clear how Equations (1) and (2) achieve the objectives of HTBA.
3. What is the difference between Equation (3) and Equation (1)? What is the loss used for $\mathcal{D}\setminus\mathcal{D_s}$, are they excluded from the training? What is Equation (4) applied on, $\mathcal{D}$ or $\mathcal{D}\setminus\mathcal{D_s}$? How Equations (3) and (4) are combined in the overall training loss?
4. Algorithm 1, “update $\delta$” and “update θ” are confusing, following which/what Equation? Where is $\mathcal{D}$ or $\mathcal{D_s}$. Line 2, “randomly initialize”, how? Line 6, retrain for how many iterations?
5. Table 1 and 2: the meaning of rows “Poisoned model patched source val (%)” is not clear. What is the size of the “source val” or “patched source val”? If their sizes are different from the “val”, then the results are not comparable to “val”, making the overall interpretation confusing.
6. In Table 4, the attack success rate under the black-box setting is rather low. So it is hard to say whether the proposed attack has achieved the claimed goals. Also the ImageNet ASR is also very low in Table 5.
7. The proposed attack is poisoning based. It should be tested against the latest training-time defense method, like ABL [1]. Also, it needs to test against recent post-training defenses like ANP [2].
8. Typo: line 127, We "stress" that.

**Limitations:**

The authors have adequately addressed the limitations and potential negative societal impact of their work.

**Strengths And Weaknesses:**

Strengths:
1. An interesting attack, i.e., HTBA, under a more realistic threat model than existing attacks: small poisoning, hidden trigger, black-box (agnostic of the victim model).
2. The proposed HTBA is quite effective as empirically verified under different poisoning rates and both gray-box and black-box settings.
3. Revealing of the practical tricks for strong HTBAs, like retraining.

Weaknesses:
1. Unclear definition or explanation of HTBA. Unclear descriptions in figure/table captions.
2. Confusing notations in the Equations. Incomplete or unclear overall learning objectives, Unclear Algorithm.
3. Missing experiments against more recent defense methods.
4. Low attack success rate under the actual black-box setting, which is one claimed highlight of the proposed threat model.

---

> ### Author Response · Authors · 2022-08-02
> **Response to Reviewer vXHc - Part 1**
>
>
> Thank you for your time and feedback! We further appreciate that you think our work is “practical and interesting”, “effective”, and our proposed threat model is “more realistic than existing attacks”. We note that in addition to this response post, we’ve also made a separate general post, with several clarifications and new results inspired by your comments. Below we address the concerns, and questions you raised.
>
> > Unclear definition or explanation of HTBA
>
> At a high level, HTBA poison victim networks to misclassify (at inference time) images which have been patched with a selected trigger. We enforce the *hidden* aspect of HTBA by constraining the poison perturbations to be small in $\ell_\infty$ norm. This is in comparison to some backdoor attacks where the trigger is directly included/superimposed on training data. Our threat model is especially insidious as modern networks are often trained on a large corpus of data scraped from the web, and minimally modified poisoned samples could easily evade detection in an automated scraping regime.
>
> As with other threat models,  HTBAs do not directly encode a trigger into poisoned data. However, our threat model focuses on $\ell_\infty$ constraints for poisons, as is the gold-standard in the adversarial attack literature, while other attacks impose constraints such as minimizing LPIPS. Furthermore, in HTBAs, a single, pre-selected trigger is chosen for poison crafting, whereas other threat models might opt for a weaker instance-based trigger.
>
> We have added some clarification in section 3.1. Important distinctions between attacks arise in the constraints on the poisoned data, and the trigger mechanism. For our attack, these are explained in section 3.2, 3.3. Does this address your question?
>
> &nbsp;
>
> > 1. Figure 1(b) is hard to understand what the columns are for. Fig. (a) also needs to include the attack step, to show the actual trigger used to mount the attack is different from the trigger patterns.
>
> Thank you for pointing out the confusion over this figure. For Figure 1b, we have clarified the source/target language in the caption. We have also replaced Figure 1a to make it more legible. 1a is a high-level schematic and doesn’t include any specifics about the attack steps, other than the addition of the crafted $\ell_\infty$ perturbation, $\delta$ to a training image, and then a *patched* inference time image. Note though that each image in this figure is taken from an actual backdoor poisoning run.
>
> Additionally, we have increased the font for Figure 1. Do you have specific questions about table captions? Does this address your concern?
>
> &nbsp;
>
> > 2. In Equation (1), the definition of $p$ (and how it relates to $\delta$),  (and $\mathcal{D}$ its difference to $\mathcal{T}$) should be clearly stated. It is not so clear how Equations (1) and (2) achieve the objectives of HTBA.
>
> We have added a clarification as to the relation of $\mathcal{T}$ and $\mathcal{D}$, and clarified $p$ for our attack. Put simply, $p$ is a trigger which will be added to images at inference time. $\delta$ is a perturbation (bounded in $\ell_\infty$) that is added to *training* data to induce misclassification on the *patched* inference time images. Equations 1,2 do not achieve the objectives of HTBA, but rather are simply the mathematical formalization of the optimization problem for HTBAs.
>
> &nbsp;
>
> > 3. What is the difference between Equation (3) and Equation (1)? What is the loss used for $\mathcal{D}$\ $\mathcal{D_s}$, are they excluded from the training? What is Equation (4) applied on, $\mathcal{D}$ or $\mathcal{D}_s$ ? How Equations (3) and (4) are combined in the overall training loss?
>
> Thank you for pointing out this confusion. Equation 1 is the outer objective of the bilevel problem introduced in Equations 1,2. Equation 3 is very related to this outer objective, but note that the adversarial loss does not explicitly depend on $\delta$. We use this adversarial loss to align the training gradients by optimizing the perturbations, $\delta$. In summary, Equation 1 is the outer part of a bilevel optimization formulation of HTBA, while Equation 3 is part of the objective that we directly *optimize* to craft the poisons (with the aim of solving the bilevel problem outlined in Equations 1,2).

---

> > ### Author Response · Authors · 2022-08-02
> > **Response to Reviewer vXHc - Part 2**
> >
> >
> > > 4. Algorithm 1, “update $\delta$” and “update $\theta$” are confusing, following which/what Equation? Where is $\mathcal{D}$ or $\mathcal{D}_s$ . Line 2, “randomly initialize”, how? Line 6, retrain for how many iterations?
> >
> > In Algorithm 1, $\delta$ is updated via signed Adam according to the alignment objective in Equation 4. Network parameters, $\theta$, are updated by retraining the crafting model on poisoned data. We simply start with training data $\mathcal{T}$, like ImageNet, so there is no sampling of data from $\mathcal{D}$. Note that the inclusion of sampling notation, and distributions, was simply to present the general and full optimization problem. In practice, our poisoning experiments are run by randomly selecting a source and target class from widely used datasets. In terms of line 2, at the beginning of crafting, perturbations are initialized from a standard normal distribution. We retrain for 80 epochs each time. For many of these details, please see Appendix C1, C2.
> >
> > &nbsp;
> >
> > > 5. Table 1 and 2: the meaning of rows “Poisoned model patched source val (%)” is not clear. What is the size of the “source val” or “patched source val”? If their sizes are different from the “val”, then the results are not comparable to “val”, making the overall interpretation confusing.
> >
> > To clarify, source data is data coming from the source class in our poisoning procedure. Data coming from this class can be patched at inference time to induce misclassification. For example, our method could be used to poison a model so that at inference time, this model mis-classifies patched images of dogs as cats. In this case, dog is the *source* class and cat is the *target* class. So source val is the validation accuracy on *clean* source images. Patched source val is the validation accuracy on *patched* source images. We apologize for any confusion.
> >
> > &nbsp;
> >
> > > 6. In Table 4, the attack success rate under the black-box setting is rather low. So it is hard to say whether the proposed attack has achieved the claimed goals. Also the ImageNet ASR is also very low in Table 5.
> >
> > To the best of our knowledge, our method achieves state of the art results in the black-box setting for our threat model. For example, even when the victim network architecture is not included in the poison crafting ensemble, our attack achieves 53% attack success - more than 10x the white-box attack success of similar attacks (Saha, Turner). For practitioners, a 53% misclassification rate on backdoored images is incredibly dangerous for industrial applications. Additionally, to our knowledge, our ImageNet results are state-of-the-art for our threat model.
> >
> > &nbsp;
> >
> > > 7. Missing experiments against more recent defense methods.
> >
> > We understand the reviewer’s concern here. However, we would like to point out that the defenses against which we deploy our attack are very highly cited and well known in the community, which is important for comparing and benchmarking attacks.
> > Nonetheless, in response to this reviewer’s concern, we have included results of our attack against the two recent defenses you cite.  This brings the total number of defenses which we test to eight.
> >
> > &nbsp;
> >
> > | Defense | Threshold | Clean model Val (%) | Attack Success Rate (%) |
> > | ----------- | ----------- | ----------- | ----------- |
> > | None   |  -   |  92.31 | 85.27 |
> > | ANP   |  0.05 |  80.05 | 51.03 |
> > | ANP   |  0.10 |  71.75 | 27.87 |
> > | ANP   |  0.15 | 50.47 | 17.77 |
> > | ANP   |  0.20 | 16.56 | 3.35 |
> >
> > &nbsp;
> >
> > | Defense | Unlearning Epochs | Clean model Val (%) | Attack Success Rate (%) |
> > | ----------- | ----------- | ----------- | ----------- |
> > | None   |  -   |  92.31 | 85.27 |
> > | ABL   |  5 | 87.53 | 70.72 |
> > | ABL   |  10 |  86.85 | 68.35|
> > | ABL   |  15 | 82.34 | 64.11 |
> > | ABL   |  20 | 64.55 | 59.51 |
> >
> >
> >
> > &nbsp;
> >
> > Thank you again for your thoughtful review. We think that your suggestions have improved our paper and introduced interesting content. We made a significant effort to address your questions, and would appreciate it if you would consider raising your score in light of our response. Please let us know if you have any additional questions we can address.

---

> > > ### Comment · Reviewer_vXHc · 2022-08-05
> > > **Thanks for the response**
> > >
> > > Thanks for the response. My concerns have been properly addressed. I am happy to increase the score to 5.

---

### Official Review · Reviewer_JjFX · 2022-07-03

**Rating:** 7
**Confidence:** 5
**Soundness:** 3 good
**Presentation:** 3 good
**Contribution:** 3 good

**Summary:**

This paper proposes an invisible poisoning-based clean-label backdoor attack against image classifiers training from scratch. The authors develop their attack based on the Hidden Trigger Backdoor Attack, which combined the benefits of both poison-label attacks and clean-label attacks whereas is effective only in the transfer learning scenarios (rather than training from scratch). Specifically, the author formulates this problem as a bi-level optimization and solves it with the classical gradient alignment technique. To ensure the generalization of the proposed attack (towards different parameter initializations and model structures), the author introduces four techniques, including 1) ensemble, 2) retraining, 3) random patch, and 4) poison selection. The author also propose a poison selection module to further enhance the attack success rate. The proposed method is tested on both CIFAR-10 and ImageNet datasets with different DNNs.

**Questions:**

1. Please provide more details about the results on Table 7. It seems that there are some detection-based methods (e.g., SS, AC, STRIP). How ASRs and BAs are calculated under these detection-based backdoor defenses?
2. The authors should review more advanced backdoor attacks (e.g., [1-3]) in Related Works.
3. Ensemble techniques have been widely used to increase the transferability among model structures, especially in adversarial learning. I think the authors should cite a few related work in Page 6 (Line 210-214) to mention it.
4. Please increase the font size in Figure 1 (a).
5. Please recheck and cite the official version of all references (e.g., [4-8]).

**Limitations:**

Limited but sufficient.

**Strengths And Weaknesses:**

Pros
1. The topic is of sufficient significance and interest to NeurIPS audiences.
2. The paper is well written and easy to follow.
3. Technically, the proposed method is moderately novel. In practice, the invisible poisoning-based clean-label backdoor attack is one of the hardest yet important problems in backdoor learning. To the best of my knowledge, there is still no work that is effective especially on large-scale datasets (e.g., ImageNet).
4. The author have also discussed the resistance to existing backdoor defenses, which should be encouraged.


In general, I enjoy the reading of this paper and recognize its significance and contributions. However, I still have some concerns about this paper. I will increase my score if the author can (partly) address my concerns. The detailed comments are as follows:


Comments
1. Please provide more details about the results on Table 7. It seems that there are some detection-based methods (e.g., SS, AC, STRIP). How ASRs and BAs are calculated under these detection-based backdoor defenses?
2. The authors should review more advanced backdoor attacks (e.g., [1-3]) in Related Works.
3. Ensemble techniques have been widely used to increase the transferability among model structures, especially in adversarial learning. I think the authors should cite a few related work in Page 6 (Line 210-214) to mention it.
4. Please increase the font size in Figure 1 (a).
5. Please recheck and cite the official version of all references (e.g., [4-8]).

References
1. Backdoor Attacks Against Deep Learning Systems in the Physical World. CVPR, 2021.
2. Invisible Backdoor Attack with Sample-Specific Triggers. ICCV, 2021.
3. WaNet - Imperceptible Warping-based Backdoor Attack. ICLR, 2021.
4. Poisoning attacks against support vector machines. ICML, 2012.
5. Witches' Brew: Industrial Scale Data Poisoning via Gradient Matching. ICLR, 2021.
6. Dataset security for machine learning: Data poisoning, backdoor attacks, and defenses. IEEE TPAMI, 2022.
7. Badnets: Evaluating backdooring attacks on deep neural networks. IEEE Access, 2019.
8. Backdoor learning: A survey. IEEE TNNLS, 2022.

---

> ### Author Response · Authors · 2022-08-02
> **Response to Reviewer JjFX**
>
> Thank you for your time and thoughtful comments! We further appreciate that you think our work is “significant and interesting”, and the writing is “easy to follow”. We note that in addition to this response post, we’ve also made a separate general post, with several clarifications and new results. Below we respond to each of your points:
>
> &nbsp;
>
> > 1. Please provide more details about the results on Table 7 ...
>
> For filtering-based defenses, such as SS, AC, we craft the poisons as usual (according to Algorithm 1). We then train a model on the poisoned data. After this, we apply one of the selected defenses to identify what training data may have been poisoned. We then remove the detected samples, and retrain a *second* network from scratch on the remaining data. Finally, we evaluate the attack success rate (on the backdoored class) using this second network. For STRIP, we simply apply the defense at test time for the first network, and filter out any patched images that exceed an entropy threshold in their predictions. In this case, an attack is considered a success if a backdoored input is not detected at test time, *and* misclassified as the target class. Thank you for pointing out this confusion, and we have since clarified this in Appendix C.4.
>
> &nbsp;
>
> > 2. The authors should review more advanced backdoor attacks (e.g., [1-3]) in Related Works.
>
> Thank you for bringing this to our attention. We have included references to the relevant works in our updated manuscript.
>
> &nbsp;
>
> > 3. Ensemble techniques ... authors should cite a few related work in Page 6 (Line 210-214) to mention it.
>
> Thank you for this suggestion. Indeed, ensembling has been used in several previous poisoning works. We have added citations to the suggested part of the manuscript.
>
> &nbsp;
>
> > 4. Please increase the font size in Figure 1 (a).
>
> We have increased the font of the descriptions. Please let us know if this is not sufficient.
>
> &nbsp;
>
> > 5. Please recheck and cite the official version of all references (e.g., [4-8]).
>
> Thank you for bringing this to our attention. We have updated references for entries about which we found additional information.
>
>
> &nbsp;
>
> Thank you again for your thoughtful review. We think that your suggestions have improved our paper content and would appreciate it if you would consider raising your score in light of our response. Please let us know if you have any additional questions we can address.

---

> > ### Comment · Reviewer_JjFX · 2022-08-03
> > **Post-rebuttal Comments**
> >
> > Thank you for the detailed responses. The authors have comprehensively addressed all my concerns. As such, I vote for acceptance.

---

### Official Review · Reviewer_npt8 · 2022-07-11

**Rating:** 5
**Confidence:** 5
**Soundness:** 2 fair
**Presentation:** 3 good
**Contribution:** 3 good

**Summary:**

In this paper, the authors propose a novel backdoor attack, Sleeper agent, that can be applied in the black-box, clean-label settings. The authors craft poisoned samples by calculating the gradient match for loss functions on the training set and patched samples. Then the authors evaluate Sleeper agent on CIFAR-10 and ImageNet two tasks using ResNet-18, MobileNet-V2, VGG11 these architectures. Finally, the authors also evaluate the robustness of Sleeper agent against several popular defense approaches. The results show that Sleeper agent can achieve high attack success rate (>80%) with a poison rate larger than 1% and perform robustness against evaluated defense mechanisms.

**Questions:**

1. How about the efficacy of Sleeper agent when the surrogate models perform worse compared with the target models.

2.  Why do STRIP and Neural Cleanse fail against Sleeper agent? I think both STRIP and Neural Cleanse are proved to be effective against static backdoor triggers as Sleeper agent.

**Ethics Review Area:**

["I don’t know"]

**Limitations:**

The evaluated defense approach is not state-of-art and some of them(i.e., Neural Cleanse) is ineffective for most proposed backdoor attacks. I think the authors should evaluate their approach against more effective defense approaches.

**Strengths And Weaknesses:**

####################Strengths####################

1. The problem is practical and interesting.

2. The evaluation is comprehensive in the dimensions of robustness, efficacy, and practice.

3. The overall writing is good and easy to follow.

####################Weakness####################

1. Threat model. Since the Sleeper agent requires the pre-trained model for crafting poisoned samples, the authors should mention such requirements or assumptions in the threat model.

2. Evaluation. The overall evaluation of Sleeper agent is comprehensive.  Regarding the transferability among pre-trained models,   the authors consider evaluating transferability among different architectures. However, I think it is also beneficial to evaluate the transferability of Sleeper agent under scenarios where the accuracy of the surrogate is different from the target models. Specifically, previous work[1] evaluates the transferability for poisoning attacks under a rather scenario where the accuracy of surrogate models is lower compared with the target models.  Moreover, I think the robustness evaluation is weak since the evaluated defense methods are not recently published. For example, the defense approach Neural Cleanse has been published 3 years, i think a more recent defense approach should be considered.


[1] Practical Poisoning Attacks on Neural Networks, ECCV 2020

---

> ### Author Response · Authors · 2022-08-02
> **Response to Reviewer npt8**
>
> Thank you for your time and feedback!  We further appreciate that you think our work is “practical and interesting”, “the evaluation is comprehensive“, and the writing is “easy to follow”.  We note that in addition to this response post, we’ve also made a separate general post, with several clarifications and new results inspired by your comments.  We respond to each of your points below:
>
> &nbsp;
>
> > 1. more recent defense approach should be considered.
>
> This point seemed to be your primary concern.  We have therefore run evaluations using two very recent defenses, ABL [1] and ANP [2]. We use ANP with various threshold values, and we report the accuracy of a ResNet-18 on CIFAR-10, where Sleeper Agent poisons have $\ell_{\infty}$-norm bounded above by $16/255$.  We find that ANP cannot achieve a low attack success rate without greatly reducing the model’s validation accuracy on clean data.  Similarly, we find that ABL must decrease accuracy substantially to remove the backdoor vulnerability.  Even after 20 epochs of unlearning, Sleeper Agent has nearly a 60% success rate.
>
> &nbsp;
>
> | Defense | Threshold | Clean model Val (%) | Attack Success Rate (%) |
> | ----------- | ----------- | ----------- | ----------- |
> | None   |  -   |  92.31 | 85.27 |
> | ANP   |  0.05 |  80.05 | 51.03 |
> | ANP   |  0.10 |  71.75 | 27.87 |
> | ANP   |  0.15 | 50.47 | 17.77 |
> | ANP   |  0.20 | 16.56 | 3.35 |
>
> &nbsp;
>
> | Defense | Unlearning Epochs | Clean model Val (%) | Attack Success Rate (%) |
> | ----------- | ----------- | ----------- | ----------- |
> | None   |  -   |  92.31 | 85.27 |
> | ABL   |  5 | 87.53 | 70.72 |
> | ABL   |  10 |  86.85 | 68.35|
> | ABL   |  15 | 82.34 | 64.11 |
> | ABL   |  20 | 64.55 | 59.51 |
>
> &nbsp;
>
> We have additionally updated our Appendix to include these results, and we will move them to the main body for the camera ready since an additional page will be permitted.
>
> &nbsp;
>
> > 2. scenarios where the accuracy of the surrogate is different from the target models.
>
> Prompted by your suggestion, we have now run additional experiments with a VGG-11 surrogate and ResNet-18 target. We find that poisons crafted on the inferior VGG-11 surrogate achieve a 57.47% attack success rate.  We have added this result to our updated draft and cited the paper you referenced.
>
> &nbsp;
>
> > 3. the authors should mention such requirements or assumptions in the threat model.
>
> We have added a clarification that the attacker does not require access to an already existing pre-trained model but rather the first step taken by an attacker may be to train their own surrogate, as in multiple other clean label and hidden trigger attacks, so we have updated Algorithm 1 and our description in the text.
>
>
> &nbsp;
>
> Thank you again for a thoughtful review. We made a significant effort to address your questions, and would appreciate it if you would consider raising your score in light of our response.  Please let us know if you have any additional questions we can address.
>
>
> &nbsp;
>
> References:
>
> [1] Li, Yige, et al. "Anti-backdoor learning: Training clean models on poisoned data." Advances in Neural Information Processing Systems 34 (2021): 14900-14912.
>
> [2] Wu, Dongxian, and Yisen Wang. "Adversarial neuron pruning purifies backdoored deep models." Advances in Neural Information Processing Systems 34 (2021): 16913-16925.

---

> ### Comment · Reviewer_npt8 · 2022-08-05
> **Thanks for your response**
>
> Thanks for your response to address my concern. I suggest you note the accuracy of each pre-trained model. I have increased my score to 5 accordingly.

---

### Official Review · Reviewer_tn2R · 2022-07-11

**Rating:** 8
**Confidence:** 5
**Soundness:** 4 excellent
**Presentation:** 4 excellent
**Contribution:** 4 excellent

**Summary:**

The paper proposes a clean label dataset poisoning attack to embed backdoors into deep neural network image classifiers. Their approach is based on training many surrogates models for generating poisoned examples and using gradient matching to find malicious perturbations. The attack, Sleeper Agent, assumes full access to the defender's training dataset which allows them to (i) train surrogate models and (ii) select which inputs to poison. Their attack requires no knowledge of the defender's victim model (or the model's architecture). The authors show that their results greatly outperform existing clean label poisoning attacks in the attack success rate and that their backdoor cannot easily be removed with existing backdoor removal or dataset augmentation methods.

**Questions:**

I wonder how much the attack's success rate depends on the attacker having access to the same training dataset as the defender. It would be interesting to relax the assumption about the overlap between the defender's and attacker's training data and see if the gradient matching still remains effective.

Do the authors have results or insights on the effectiveness of their Sleeper Agent attack when the defender fine-tunes uses a pre-trained model?

**Limitations:**

-

**Strengths And Weaknesses:**

** Strengths

* High Relevance and Effectiveness. Poisoning attacks with few assumptions about the victim's model architecture or training procedure are highly relevant. The paper shows clear improvement over existing works in this field and demonstrates that their attack has practical relevance.

* Ablation studies. The experiments are convincing and show that the author's approach has high transferability of the backdoor across model architectures and datasets (extension to ImageNet). The authors also show a positive effect on the attack success rate if the attacker is allowed to use larger perturbations.

* Paper Organization. The paper is neatly organized and shows all results in a comprehensive manner. Discussed ideas were easy to follow and provided comprehensive intuition and motivation.

** Weaknesses

Overall, the paper was an interesting read and convinced me of the effectiveness of the author's attack. I believe the paper is of high interest to the research community.

* Limited defense evaluation. Since the generated patterns resemble an adversarial example, it seems plausible to evaluate defenses that harden the model against adversarial attacks, such as adversarial training during training.

* Missing Runtime on ImageNet. Figure 5 shows a high runtime for CIFAR-10, but it appears that the runtime for ImageNet is missing.

---

> ### Author Response · Authors · 2022-08-02
> **Response to Reviewer tn2R**
>
> Thank you for your time and encouraging feedback!  We further appreciate that you think our work is “highly relevant”, constitutes “clear improvement over existing works in this field”, contains “convincing” experiments, and is “easy to follow”.  We note that in addition to this response post, we’ve also made a separate general post, with several clarifications and new results inspired by your comments.  We respond to each of your points below:
>
> &nbsp;
>
> > 1. defense evaluation
>
> Prompted by your review, we have run several additional defenses including your suggestion, adversarial training.  We also evaluate against the ANP defense with various threshold values, and we report the accuracy of a ResNet-18 on CIFAR-10, where Sleeper Agent poisons have $\ell_{\infty}$-norm bounded above by $16/255$.  We find that ANP cannot achieve a low attack success rate without greatly reducing the model’s validation accuracy on clean data.  Similarly, we find that ABL must decrease accuracy substantially to remove the backdoor vulnerability.  Even after 20 epochs of unlearning, Sleeper Agent has nearly a 60% success rate.  Across all three new defenses, the only case where the defender reduces to ASR to a low success rate is ANP with a high threshold such that clean accuracy is unacceptably low.
>
> &nbsp;
>
> | Defense | Threshold | Clean model Val (%) | Attack Success Rate (%) |
> | ----------- | ----------- | ----------- | ----------- |
> | None   |  -   |  92.31 | 85.27 |
> | ANP   |  0.05 |  80.05 | 51.03 |
> | ANP   |  0.10 |  71.75 | 27.87 |
> | ANP   |  0.15 | 50.47 | 17.77 |
> | ANP   |  0.20 | 16.56 | 3.35 |
>
> &nbsp;
>
> | Defense | Unlearning Epochs | Clean model Val (%) | Attack Success Rate (%) |
> | ----------- | ----------- | ----------- | ----------- |
> | None   |  -   |  92.31 | 85.27 |
> | ABL   |  5 | 87.53 | 70.72 |
> | ABL   |  10 |  86.85 | 68.35|
> | ABL   |  15 | 82.34 | 64.11 |
> | ABL   |  20 | 64.55 | 59.51 |
>
> &nbsp;
>
> | Defense | Clean model Val (%) | Attack Success Rate (%) |
> | ----------- | ----------- | ----------- |
> | None   | 92.31 | 85.27 |
> | Adv. Training   |  88.63 | 52.83 |
>
> &nbsp;
>
> We have additionally updated our Appendix to include these results, and we will move them to the main body for the camera ready since an additional page will be permitted.
>
> &nbsp;
>
> > 2. Runtime on ImageNet
>
> We have now added these to our updated draft in Appendix C.5.  Thank you for pointing out that these were missing.
>
> &nbsp;
>
> > 3. how much the attack's success rate depends on… access to the same training dataset...
>
> This is an interesting point and one that is worth posing to the community at large!  Other attacks such as Hidden Trigger Backdoor Attacks and Clean-Label Backdoor Attacks also have this requirement as well as targeted data poisoning attacks like Poison Frogs, Meta-Poison, Witches’ Brew, etc.  In order to expand the applicability of such attacks, rethinking this assumption is important.  We have now run a new experiment in which the attacker sees half of the CIFAR-10 training set when crafting poisons, and the victim then trains on the other half of the CIFAR-10 training set along with the poisons.  With ResNet-18 victim and 1% poison budget (250 images in this setting), Sleeper Agent achieves a 66.27% success rate, even though the victim trains on data unknown to the attacker.  We are now running this experiment on additional splits of the training data and will add these to the camera ready version.
>
> &nbsp;
>
> > 4. defender fine-tunes uses a pre-trained model?
>
> Inspired by your feedback, we have now run an experiment where the defender fine-tunes a torchvision ResNet-18 model on CIFAR-10.  With a 1% poison budget, the attacker still achieves a 55.81% success rate despite being completely unaware of the victim’s pre-trained model.  We will similarly run a sweep across torchvision pre-trained models and include these in our updated draft.
>
> &nbsp;
>
> Thank you again for your thoughtful and supportive review. We think that your suggestions have improved our paper and introduced interesting content.  We made a significant effort to address your questions, and would appreciate it if you would consider raising your score in light of our response.  Please let us know if you have any additional questions we can address.
>
> &nbsp;
>
> References:
>
> [1] Li, Yige, et al. "Anti-backdoor learning: Training clean models on poisoned data." Advances in Neural Information Processing Systems 34 (2021): 14900-14912.
>
> [2] Wu, Dongxian, and Yisen Wang. "Adversarial neuron pruning purifies backdoored deep models." Advances in Neural Information Processing Systems 34 (2021): 16913-16925.

---

### Author Response · Authors · 2022-08-02
**General Response to Reviewers**

We thank the reviewers for their feedback and support. We here provide a general response, addressed to all reviewers and ACs, as well as individual replies to address specific reviewer points as separate posts.  As noted by the reviewers, our work addresses a relevant and well-motivated problem of interest to the NeurIPS audience, and our proposed backdoor attack demonstrates “clear improvement over existing works” across comprehensive evaluations that include hard problems like ImageNet.  Prompted by feedback from the reviewers, we have now run a number of new experiments.  We present a subset of these experiments below along with a discussion of two points noted by reviewers:

Evaluation against recent defenses: A common thread amongst reviews was a request to analyze additional recent defenses.  In addition to the 6 defenses already evaluated in the original submission, we have now run two recent defenses, ABL [1] and ANP [2]. We evaluate against ANP with various threshold values, and we report the accuracy of a ResNet-18 on CIFAR-10, where Sleeper Agent poisons have $\ell_{\infty}$-norm bounded above by $16/255$.  We find that ANP cannot achieve a low attack success rate without greatly reducing the model’s validation accuracy on clean data.  Similarly, we find that ABL must decrease accuracy substantially to remove the backdoor vulnerability.  Even after 20 epochs of unlearning, Sleeper Agent has nearly a 60% success rate.

&nbsp;

| Defense | Threshold | Clean model Val (%) | Attack Success Rate (%) |
| ----------- | ----------- | ----------- | ----------- |
| None   |  -   |  92.31 | 85.27 |
| ANP   |  0.05 |  80.05 | 51.03 |
| ANP   |  0.10 |  71.75 | 27.87 |
| ANP   |  0.15 | 50.47 | 17.77 |
| ANP   |  0.20 | 16.56 | 3.35 |

&nbsp;

| Defense | Unlearning Epochs | Clean model Val (%) | Attack Success Rate (%) |
| ----------- | ----------- | ----------- | ----------- |
| None   |  -   |  92.31 | 85.27 |
| ABL   |  5 | 87.53 | 70.72 |
| ABL   |  10 |  86.85 | 68.35|
| ABL   |  15 | 82.34 | 64.11 |
| ABL   |  20 | 64.55 | 59.51 |

&nbsp;

Clarifying notation and assumptions:  Two reviewers have pointed out that notation and assumptions made by the attacker could use further discussion in the paper.  We have thus updated sections 1, 2, and 3 of our draft to reflect these clarifications.

References:

[1] Li, Yige, et al. "Anti-backdoor learning: Training clean models on poisoned data." Advances in Neural Information Processing Systems 34 (2021): 14900-14912.

[2] Wu, Dongxian, and Yisen Wang. "Adversarial neuron pruning purifies backdoored deep models." Advances in Neural Information Processing Systems 34 (2021): 16913-16925.

---

### Meta-Review · Area_Chair_EhLY · 2022-08-24

**Recommendation:** Accept
**Confidence:** Certain

**Metareview:**

This paper introduces a clean label poisoning attack to backdoor neural network models.
The reviewers unanimously voted to accept this paper and I agree: it is a strong
technical paper that is well written. The experiments were initially limited in some
areas but the author's rebuttal addresses many of these concerns.


**Award:**

No

---

### Decision · Program_Chairs · 2022-09-14

Accept